# Functional cardiac fibroblasts derived from human pluripotent stem cells via second heart field progenitors

Jianhua Zhang[1,2], Ran Tao[1], Katherine F. Campbell[3,4], Juliana L. Carvalho[1,5], Edward C. Ruiz[1], Gina C. Kim[1], Eric G. Schmuck[1], Amish N. Raval[1], André Monteiro da Rocha[3,4], Todd J. Herron[3,4], José Jalife[3,6], James A. Thomson[7,8] & Timothy J. Kamp [1,2,8]

Cardiac fibroblasts (CFs) play critical roles in heart development, homeostasis, and disease. The limited availability of human CFs from native heart impedes investigations of CF biology and their role in disease. Human pluripotent stem cells (hPSCs) provide a highly renewable and genetically defined cell source, but efficient methods to generate CFs from hPSCs have not been described. Here, we show differentiation of hPSCs using sequential modulation of Wnt and FGF signaling to generate second heart field progenitors that efficiently give rise to hPSC-CFs. The hPSC-CFs resemble native heart CFs in cell morphology, proliferation, gene expression, fibroblast marker expression, production of extracellular matrix and myofibroblast transformation induced by TGFβ1 and angiotensin II. Furthermore, hPSC-CFs exhibit a more embryonic phenotype when compared to fetal and adult primary human CFs. Co-culture of hPSC-CFs with hPSC-derived cardiomyocytes distinctly alters the electrophysiological properties of the cardiomyocytes compared to co-culture with dermal fibroblasts. The hPSC-CFs provide a powerful cell source for research, drug discovery, precision medicine, and therapeutic applications in cardiac regeneration.

[1] Department of Medicine, School of Medicine and Public Health, University of Wisconsin-Madison, Madison, WI 53705, USA. [2] Stem Cell and Regenerative Medicine Center, University of Wisconsin-Madison, Madison, WI 53705, USA. [3] Center for Arrhythmia Research, Department of Internal Medicine, Cardiovascular Medicine, University of Michigan, Ann Arbor, MI 48109, USA. [4] Frankel Cardiovascular Regeneration Core Laboratory, Ann Arbor, MI 48109, USA. [5] Department of Genomic Sciences and Biotechnology, Catholic University of Brasilia, Brasilia 70790 Distrito Federal, Brazil. [6] Fundación Nacional Centro Nacional de Investigaciones Cardiovasculares Carlos III (CNIC), Melchor Fernández Almagro, 328029 Madrid, Spain. [7] Regenerative Biology Division, Morgridge Institute for Research, Madison, WI 53715, USA. [8] Department of Cell and Regenerative Biology, School of Medicine and Public Health, University of Wisconsin-Madison, Madison, WI 53705, USA. Correspondence and requests for materials should be addressed to J.Z. (email: jz2@medicine.wisc.edu) or to T.J.K. (email: tjk@medicine.wisc.edu)

Cardiac fibroblasts (CFs) compromise a significant fraction of cells in the heart estimated using lineage tracing to be about 15% of the nonmyocyte cells in mouse heart[1], and CFs contribute to heart function, homeostasis, and structure in multiple ways. The production and remodeling of extracellular matrix (ECM) are the best known functions of CFs. Secretion of growth factors and complex mixtures of regulatory molecules in extracellular vesicles provide another important way that CFs regulate cardiac function. During development, signaling from CFs is necessary to stimulate cardiomyocyte proliferation[2]. CFs are intimately involved in many of the dysregulated signaling pathways linked to heart disease such as β-adrenergic signaling in heart failure[3]. Activated CFs and resulting pathological fibrosis can lead to impaired mechanical function associated with heart failure as well as abnormal electrical activity linked to arrhythmias and sudden cardiac death[4]. Not surprisingly, with this range of cardiac-specific effects, evidence has emerged that CFs are tissue-specific cells distinct from other fibroblasts in the body[5].

Given the role of CFs in cardiac biology and disease, investigators have isolated and studied primary CFs from surplus patient material obtained at the time of cardiac surgery, cardiac biopsy, or autopsy. However, disease- and patient-specific cardiac tissue samples are not always available. For example, many patients with inherited heart disease typically do not undergo cardiac surgical interventions. Thus, a robust, reproducible, and patient-specific source of CFs is desirable. Because human pluripotent stem cells (hPSCs) can renew indefinitely in culture, hPSCs provide an unlimited source of genetically identical hPSC-CFs in contrast to primary tissue sources of CFs. Prior studies have identified fibroblasts as a low-abundance cell type present following protocols optimized to differentiate hPSCs to cardiomyocytes (hPSC-CMs)[6,7]. In addition, recent investigations have defined protocols to generate epicardial cells from hPSCs that subsequently can be differentiated into CFs[8–11]. However, efficient differentiation of hPSCs to well characterized CFs has not been described.

During embryonic development, CFs arise from multiple sources. The epicardium is considered the major source of CFs found in the ventricular myocardium[12–16] with epicardial cells undergoing an epithelial-to-mesenchymal transition requiring expression of Tcf21 to generate CFs[12,13,17,18]. Lineage tracing with the epicardial marker, Tbx18, demonstrated Tbx18-expressing fibroblasts compromised only one-third of the CFs in embryonic and adult heart[13]; however, a more recent investigation with a Wt1-Cre mouse showed up to 80% of CFs in the adult heart were derived from the epicardium[19]. Another significant population of CFs is derived from the endocardium at the time of endocardial cushion formation by an endothelial-to-mesenchymal transition[19,20], and endocardial cells are generated in part from the second heart field progenitors (SHFPs)[21]. Lastly, a small fraction of fibroblasts are derived from neural crest lineages[22].

In the present study, we develop a protocol to generate hPSC-CFs via SHFPs using sequential inhibition of GSK3β to activate canonical Wnt signaling followed by stimulation with bFGF. The differentiated hPSC-CFs exhibit cell morphology, growth, gene expression, fibroblast markers, ECM production, and myofibroblast transformation similar to native human CFs.

## Results

**Progenitors arise during cardiac differentiation of hPSCs.** Biphasic modulation of the Wnt signaling pathway with small molecules is sufficient to direct the differentiation of monolayer cultured hPSCs to CMs[23,24]. Activation of canonical Wnt signaling by GSK3β inhibition stimulates hPSCs to sequentially form mesoderm and cardiac progenitors that are subsequently induced to differentiate to CMs by inhibition of Wnt signaling. We first determined the stage-specific mesodermal and cardiac progenitor populations generated by this small molecule (GiWi) protocol (Fig. 1a). Flow cytometry demonstrated that the early mesodermal marker Brachyury (Bry) was expressed in almost all cells after 24 h of treatment with CHIR. Expression of Bry was present for another 24 h before it was downregulated after day 2 (Fig. 1b, Supplementary Fig. 1a). CD90 was robustly expressed in hPSCs, but its expression gradually declined during the differentiation. Most of the cells expressed CD90 at days 2–3 (>80%), but the expression level of CD90 was lower than in hPSCs (Fig. 1b, Supplementary Fig. 1b). This Bry$^{down}$/CD90$^{low}$ progenitor stage at day 2–3 first showed upregulation of *MESP1* and *GATA4* mRNA expression which are expressed in cardiac mesodermal progenitors (Fig. 1c), followed by the upregulation of cardiac transcription factors *ISL1*, *NKX2-5*, and *TBX5* indicating commitment of cardiac progenitors in the GiWi protocol (Fig. 1c). The apelin receptor (APLNR) is expressed in mesodermal progenitors including lateral plate mesodermal cells specified to be cardiac progenitors as well as APLNR$^+$ cells that have the potential to give rise to mesenchymal stem cells (MSCs) and endothelial cells[25–28]. APLNR expression was first seen at day 3, and APLNR$^+$ cells peaked at 66% of the cells on day 4 and then rapidly declined (Fig. 1b, Supplementary Fig. 1c). KDR$^+$/PDGFRα$^+$ cells have been identified as cardiac progenitor cells (CPCs) that can be differentiated mainly to CMs in the cardiac differentiation of hPSCs[6]. We found the KDR$^+$/PDGFRα$^+$ CPCs were mainly generated on day 4–5 (Fig. 1b, Supplementary Fig. 1d). These stage-specific progenitors were reproducibly generated from other hPSC lines using the GiWi protocol (Supplementary Fig. 2).

**FGF promotes differentiation of SHFPs and CFs.** Because prior studies have demonstrated that fibroblast growth factor (FGF) signaling contributes to the generation of cardiac mesodermal progenitors and exerts a dominant effect over BMP signaling directing cardiac mesodermal progenitors to nonmyocyte fates[29,30], we tested the impact of FGF signaling on the stage-specific progenitors present in the GiWi protocol to promote differentiation of CFs rather than CMs (Fig. 2a). We examined a range of concentrations of bFGF (0–125 ng/ml) added at different cardiac progenitor stages in a defined medium (CFBM, Supplementary Table 1) for fibroblast differentiation. Following 20 days of differentiation, cells were analyzed by flow cytometry using a specific fibroblast antibody (mouse anti-human fibroblast, clone TE-7[31]) and an antibody to sarcomeric myosin (MF20) expressed in CMs (Fig. 2b, Supplementary Fig. 3). Changing the medium and adding bFGF on day 2 or day 3 effectively prevented the generation of MF20$^+$ CMs, and there was a concentration-dependent increase in fibroblast marker positive cells, which peaked at 75 ng/ml bFGF added on day 2 of differentiation (Fig. 2b). Higher concentrations of bFGF, when added on day 3 or day 4, resulted in significant cell death after day 6 of differentiation and no cells survived to 20 days (Fig. 2b). Applying a fixed concentration of 75 ng/ml of bFGF to cardiac mesodermal progenitors in a context-dependent fashion generated large population of fibroblasts from multiple hPSC lines which we refer to as hPSC-CFs (Fig. 2c).

We focused our effort on the protocol adding 75 ng/ml of bFGF in CFBM medium starting on day 2 following GSK3β inhibition (Gi) which we describe as the GiFGF protocol (Protocol I in Fig. 2a). To understand the impact of bFGF on the differentiation of day 2 cardiac mesodermal progenitors in the GiFGF protocol, we examined the temporal gene-expression

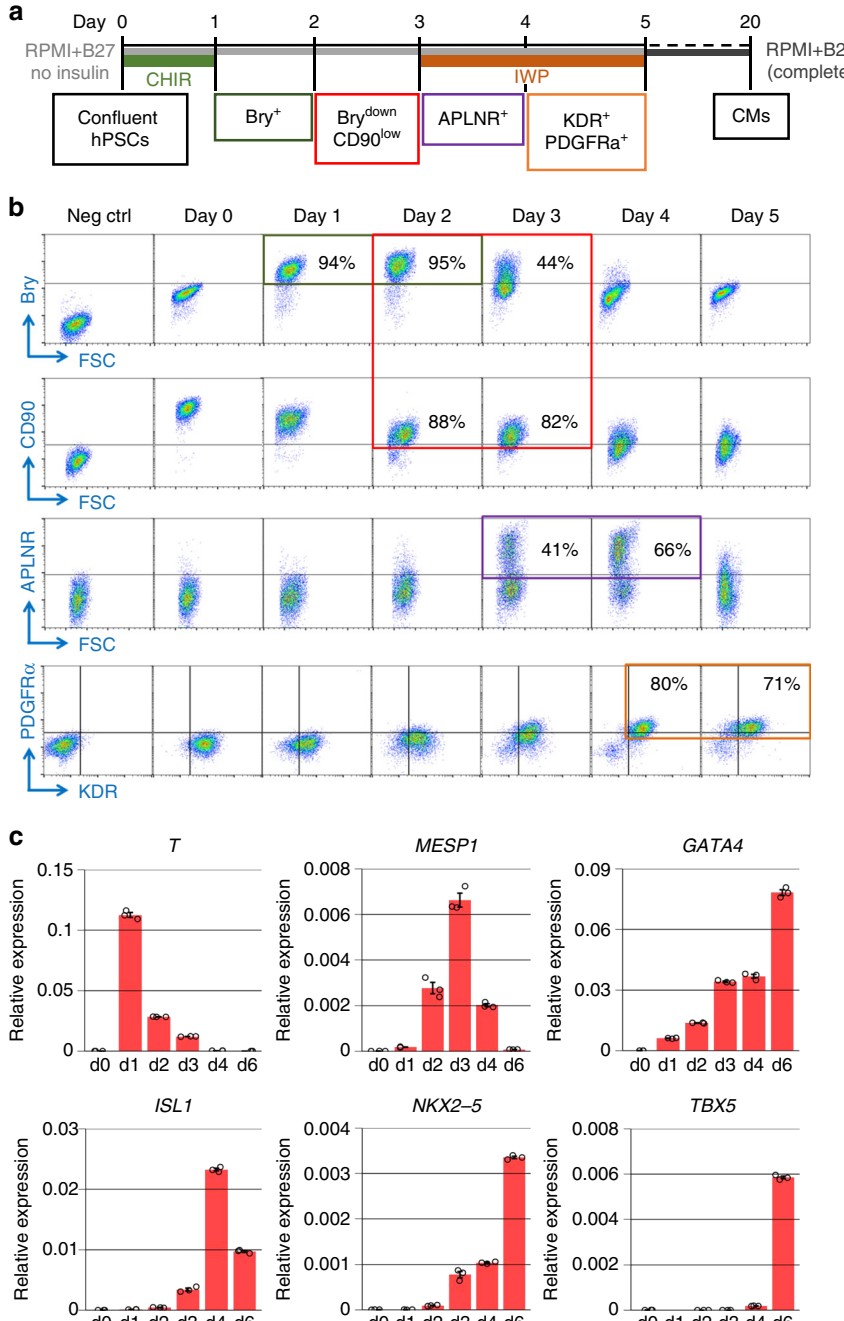

**Fig. 1** Identification of progenitors in cardiac differentiation of hPSCs. **a** Schematic method for the small molecule protocol using GSK3β inhibition with CHIR followed by Wnt inhibition with IWP (GiWi protocol) to efficiently differentiate hPSCs to cardiomyocytes (CMs) and the associated markers for stage-specific progenitors. **b** Flow cytometry of stage-specific progenitors labeled by Brachyury (Bry), CD90, Apelin receptor (APLNR), KDR, and PDGFRα in early differentiation (day 0–5) of the GiWi protocol. No primary antibody controls and isotype controls were performed for each time point, and the day 0, no primary antibody control (Neg ctrl) is shown as an example. **c** qRT-PCR showing the expression of relevant mesodermal and cardiac-related transcription factors in the progenitor stages of the GiWi protocol (day 0–6, n = 3 technical replicates). *TBX5*, d1 expression was not detectable. Data are mean ± SEM. The data are from DF19-9-11T hiPSC line

pattern for developmentally regulated transcription factors and associated genes using quantitative real time polymerase chain reaction (qRT-PCR) (Fig. 3a, Supplementary Fig. 4). The early mesodermal gene *T* was transiently upregulated after the GSK3β inhibitor (CHIR) treatment peaking at day 1. *MESP1* and *GATA4* started to express on day 2 of differentiation following the expression of *T*, indicating the formation of cardiac mesodermal progenitors (Fig. 3a). Next, the cardiac-related transcription factors were upregulated following bFGF treatment including

*ISL1*, *TBX1*, *GATA4*, *NKX2-5*, *HAND2* (Fig. 3a, Supplementary Fig. 4). The expression of *ISL1*, *GATA4*, and *HAND2* remained high through 20 days of differentiation. Interestingly, the pattern of expression is consistent with the formation of SHFPs given the prominent expression of *ISL1*, *TBX1*, *GATA4*, and *HAND2* (Fig. 3b)[32–36]. The results are also consistent with the demonstrated role of FGF signaling driving differentiation of pharyngeal mesoderm to SHFPs[37]. These results contrast the cardiomyocyte-optimized GiWi protocol in which transcription factors

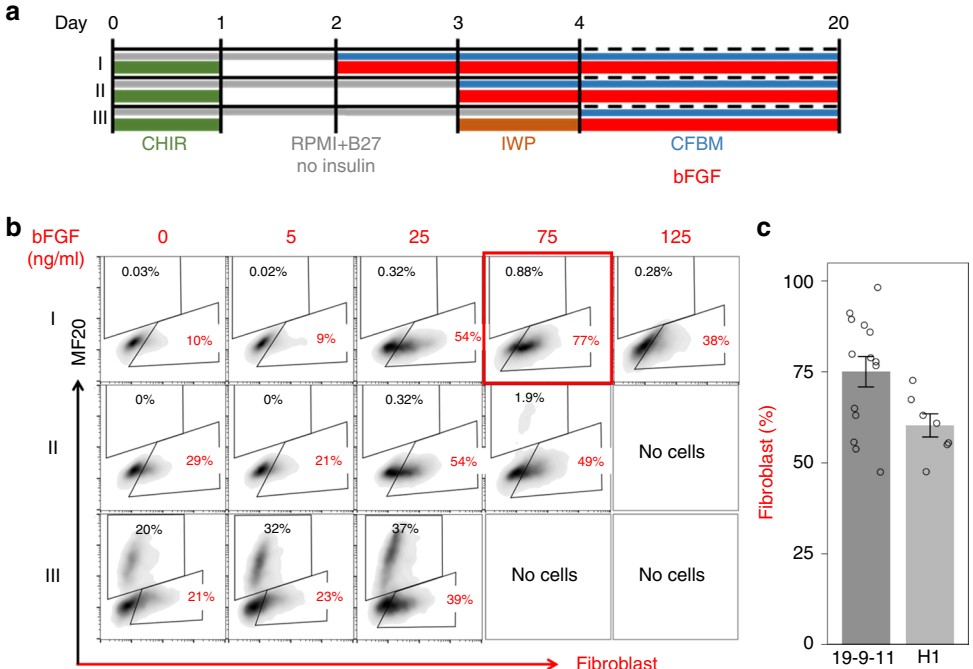

**Fig. 2** Effect of bFGF on hPSC-derived mesodermal and cardiac progenitors. **a** Schematic for testing the concentration-dependent effect of bFGF addition to stage-specific progenitors generated by the GiWi protocol beginning bFGF on day 2, 3, or 4, corresponding to labeled protocols I, II, and III. Gray lines indicate RPMI medium+B27 without insulin supplement; blue lines indicate cardiac fibroblast basal medium (CFBM); green lines indicate CHIR treatment; red lines indicate bFGF treatment; orange line indicates IWP treatment. **b** Flow cytometry analysis of day 20 differentiated cells from protocols I, II, III labeled by human fibroblast marker (clone TE-7 antibody) and cardiomyocyte marker (MF20 antibody for sarcomeric myosin). **c** Average percentage of fibroblast population differentiated from hESC line H1 (n = 7, biological replicates) and hiPSC line DF19-9-11T (n = 14, biological replicates) at 20 days. Data are mean ± SEM

associated with first heart field (FHF) progenitors including *TBX5*[38], *HAND1*, and *TBX20* are more prominently expressed (Fig. 3b). In addition, the ion channel gene, *HCN4*, which is recognized as the FHF progenitor marker[39], was minimally expressed in the GiFGF protocol compared to the GiWi protocol (Fig. 3b). Furthermore, the cardiac transcription factors *ISL1*, *NKX2-5*, and *GATA4*, which are expressed in both FHF and SHF, were differentially expressed in the GiFGF and GiWi protocols (Fig. 3b). Particularly, *ISL1* expression persisted longer in GiFGF protocol compared to an early peak in expression in the GiWi protocol on day 4 then declining. In contrast, *NKX2–5* expression peaked early in the GiFGF protocol at day 6 and rapidly downregulated after day 10 compared to the GiWi protocol where *NKX2-5* expression increased after day 10 and is greatest at day 20. *TCF21*, a bHLH transcription factor required for the formation of CF in mouse heart[18], was progressively upregulated from day 6–20 in the GiFGF protocol (Fig. 3a, Supplementary Fig. 4). However, we did not observe a significant upregulation of (pro)epicardial genes such as *TBX18* and *WT1* with the FGF directed CF differentiation (Fig. 3a, Supplementary Fig. 4), consistent with the findings that FGF signaling inhibits (pro) epicardium differentiation from the cardiac mesoderm[29]. We also examined the EMT markers of *SNAI1* and *SNAI2* in the GiFGF protocol and found an early upregulation of *SNAI1* (day 2–3) and a late upregulation of *SNAI2* (day 6–20) (Fig. 3a). *THY1* (CD90) expression was high in the undifferentiated hPSCs but downregulated during CF differentiation (Fig. 3c). Given that high bFGF concentrations can be supportive of maintenance of pluripotency of hPSCs, we examined the expression of the pluripotency gene *OCT4* and showed it to be completely downregulated during the CF differentiation (Fig. 3c), similar to the downregulation observed in cardiomyocyte differentiation

protocols[7,23,40,41]. Finally, we examined the cell lineage-specific gene-expression pattern. *VIM*, commonly expressed in fibroblasts, was greatly upregulated during CF differentiation in contrast to *TNNT2*, a cardiomyocyte-specific gene, which is minimally expressed in the GiFGF protocol (Fig. 3d). Furthermore, *POSTN*, a gene expressed in CFs[42], was also greatly upregulated by 20 days differentiation in the GiFGF protocol, whereas the general fibroblast-associated *DDR2* gene showed a smaller fold change compared to *POSTN* (Fig. 3d). To screen for gene expression typical of endothelial or smooth muscle cells, we examined *PECAM1* (CD31) and *MYH11* (smooth muscle myosin heavy chain) expression and found there was very low expression of *PECAM1* and essentially no expression of *MYH11* in the GiFGF protocol (Fig. 3d).

Based on the gene-expression data showing SHF genes were significantly upregulated upon bFGF addition in the GiFGF protocol, we used flow cytometry to examine the temporal expression of Islet1 and the recently identified SHF marker CXCR4 (CD184)[43] in the GiFGF protocol compared to the GiWi protocol (Fig. 4a, Supplementary Fig. 5). Islet1 expressing cells first were detected on Day 3 of both protocols, but bFGF signaling resulted in the majority of cells being Islet1 positive in the GiFGF protocol relative to a smaller fraction in the GiWi protocol during the progenitors stages (Day 3–5). Furthermore, Islet1/CXCR4 double positive cells were the majority of cells (69%) on Day 5 in the GiFGF protocol and were only scarcely detected in the GiWi protocol. After 20 days of differentiation, we evaluated for the different cell lineages present and confirmed the GiWi protocol gives rise to predominantly hPSC-CMs (MF20+) and the GiFGF protocol gives rise to predominantly hPSC-CFs (fibroblast TE-7+) with a small population of smooth muscle cells (SM-MHC+) in the GiWi protocol only (Fig. 4b, Supplementary Fig. 6). No

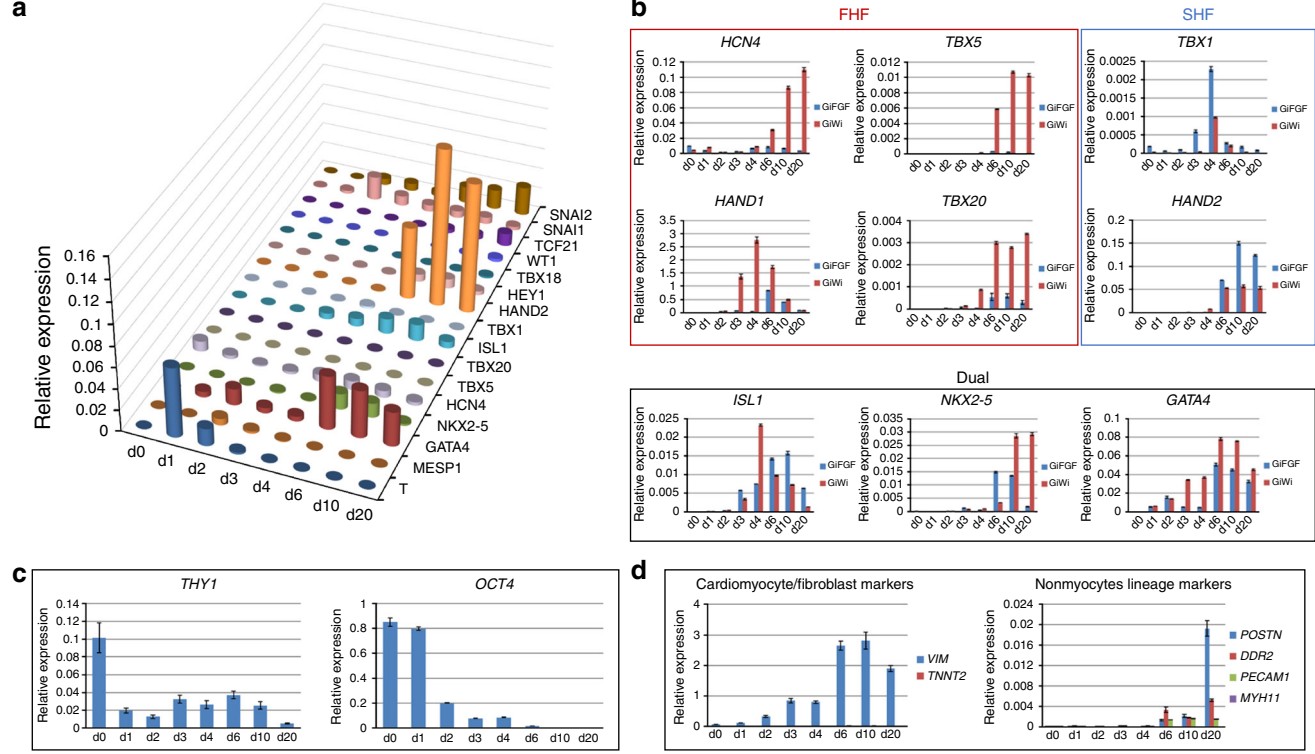

**Fig. 3** Analysis of cardiogenic gene expression during GiFGF protocol. **a** qRT-PCR showing the expression and hierarchy of transcription factors in the time course of the GiFGF protocol (day 0–20). **b** qRT-PCR showing the expression patterns of the transcription factors in FHF and SHF in the GiFGF protocol in comparison with the GiWi protocol. **c** qRT-PCR showing the expression time course of CD90 and OCT4 in GiFGF protocol (day 0–20). **d** qRT-PCR showing the expression of markers for fibroblasts (*VIM*, *POSTN*, and *DDR2*), cardiomyocytes (*TNNT2*), and nonmyocyte lineages (*PECAM1* and *MYH11*) in the time course of the GiFGF protocol (day 0–20). The qRT-PCR time course data are from consecutive samples from the same differentiation round with $n = 3$ technical replicates for each point. Data are mean ± SEM. The data are from DF19-9-11T hiPSC line

CD31[+] endothelial cells were detected under these conditions (Fig. 4b, Supplementary Fig. 6).

We sought to further define the lineage hierarchy of the hPSC-CFs by evaluating for expression of a marker associated with vascular endothelial progenitors, Tie2[44], and the endocardial-specific transcription factor during early cardiac development, NFATc1[45]. We also examined the time course of expression of Islet1 as endocardial populations have been described from Islet1-positive and -negative populations in the developing mouse heart[21]. Despite the persistent expression of Islet1 detected from day 4 to day 20 in the GiFGF protocol, no significant Tie2-positive cells were observed (Fig. 4c, Supplementary Fig. 7). Evaluation for NFATc1 expression from day 10 to day 20 also showed no significant population of NFATc1-positive cells (Supplementary Fig. 8). A time course of the generation of hPSC-CFs during the GiFGF protocol shows the appearance of TE-7[+] cells starting on day 6 and increasing in abundance to day 18 consistent with the generation of hPSC-CFs from Islet1/CXCR4 double positive SHFPs (Fig. 4d). In summary, the GiFGF protocol generates an enriched pool of SHFPs that give rise to hPSC-CFs without going through an evident endocardial progenitor cell stage (Fig. 4e).

**Comparison of hPSC-CFs, primary CFs, and dermal fibroblasts**. We next compared the properties of hPSC-CFs with commercially available primary human fetal ventricular CFs (hfV-CFs) and adult ventricular CFs (haV-CFs), and adult dermal fibroblasts (hDFs). The morphology of hPSC-CFs, hfV-CFs, and haV-CFs maintained in similar media was comparable with spindle-shaped and polygonal cells present, but distinct from

hDFs which were larger, flatter and more irregular in shape (Fig. 5a). The hPSC-CFs and hfV-CFs were more proliferative than haV-CFs and can propagate for more than 14 passages, while haV-CFs could only grow ~9 passages before undergoing senescence (Fig. 5b). The expression of CD90 (*THY1*), which has been used as a cell surface marker for some fibroblast populations, was uniformly expressed on hDFs and maintained over multiple passages. In contrast, expression of CD90 was present on only a fraction of hPSC-CFs, hfV-CFs and haV-CFs (~30%) and was present on a progressively smaller percentage of cells with passaging. Notably, the hPSC-CFs, hfV-CFs, and haV-CFs showed a similar change in expression of CD90 over passages (Fig. 5c, Supplementary Fig. 9). The hDFs were uniformly positive (~100%) for the intracellular fibroblast marker (clone TE-7) during passaging, and the hPSC-CFs, hfV-CFs, and haV-CFs showed that 80–90% of the cells were positive at early passage. The TE-7-positive cells increased to 99% with subsequent passaging and then declined as the CFs approached senescence (Fig. 5c Supplementary Fig. 10).

We tested labeling vimentin, an intermediate filament protein expressed in fibroblasts, and show that all of the clone TE-7 fibroblast antibody labeled cells from hPSC-CFs, hfV-CFs, haV-CFs, and hDFs populations also uniformly expressed vimentin, whereas hPSC-CMs did not express vimentin (Fig. 6a, Supplementary Figs. 11 and 12). The surface markers of PDGFRα (CD140a) and PDGFRβ (CD140b) have been reported to label a portion of primary mouse and human CFs[5], and so we also examined their expression in hPSC-CFs, hfV-CFs, haV-CFs, hDFs, and hPSC-CMs by flow cytometry. PDGFRα was expressed in more than 90% of the cells in all the fibroblast populations, but not in hPSC-CMs (Fig. 6b, Supplementary Fig. 13). PDGFRβ was

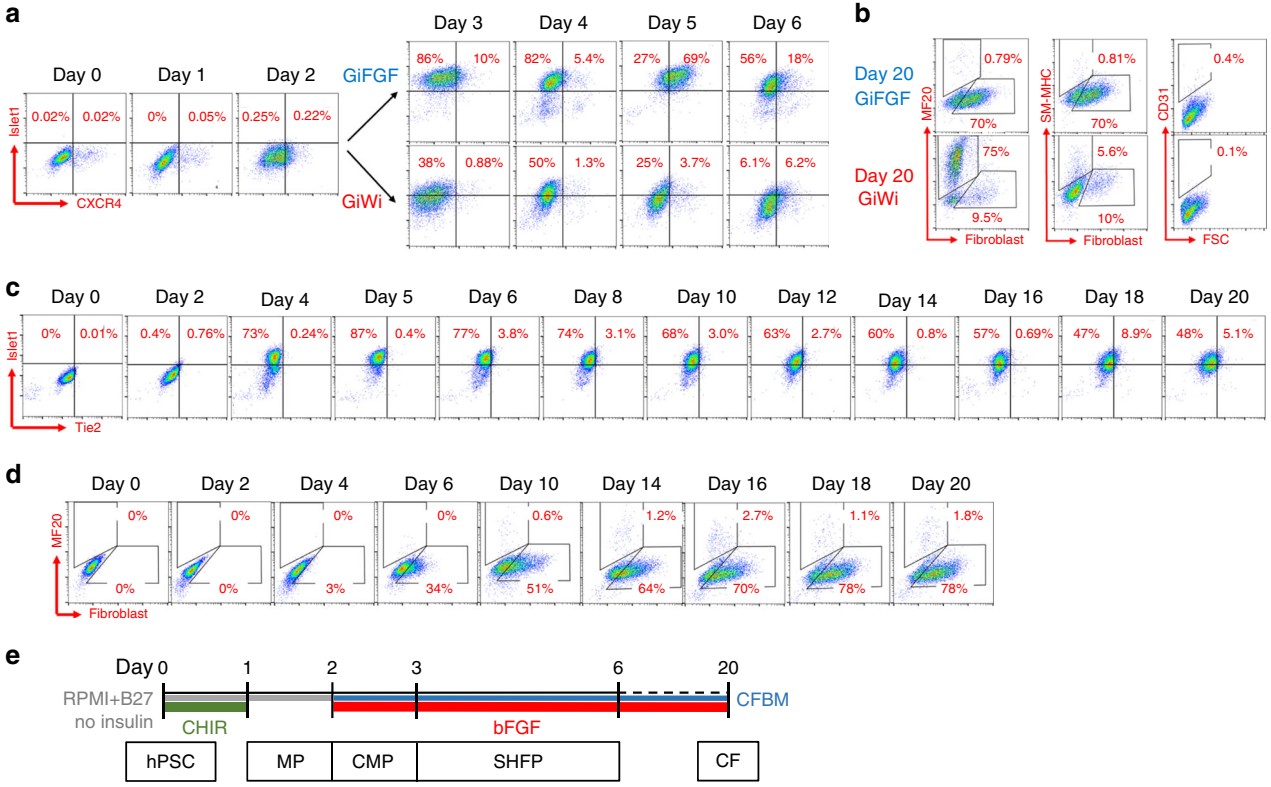

**Fig. 4** Evaluation of progenitors and cardiovascular lineages in CF and CM differentiation. **a** Flow cytometry analysis of cells on day 0–6 in the GiFGF and GiWi protocols for FHF/SHF marker Islet1 and SHF marker CXCR4. For day 0–2, the protocols are identical, diverging on day 3. **b** Flow cytometry analysis of the day 20 differentiated cells from the GiFGF and GiWi protocols for fibroblasts (anti-human fibroblasts, clone TE-7), CMs (MF20), smooth muscle cells (SM-MHC), and endothelial cells (CD31). **c** Flow cytometry analysis of cells throughout the GiFGF protocol for expression of Islet1 and the endothelial progenitor marker, Tie2. **d** Flow cytometry analysis of cells throughout the GiFGF protocol for expression of the fibroblast marker (anti-human fibroblasts, clone TE-7). **e** Schematic method of the GiFGF protocol and stage-specific progenitors in differentiation of hPSCs to CFs. Gray lines indicate RPMI medium +B27 without insulin supplement; blue lines indicate cardiac fibroblast basal medium (CFBM); green lines indicate CHIR treatment; red lines indicate bFGF treatment. MP mesodermal progenitor, CMP cardiac mesodermal progenitor, SHFP second heart field progenitor, CF cardiac fibroblast. The data are from DF19-9-11T hiPSC line

also expressed in the majority of the hPSC-CFs and hfV-CFs, as wells as hDFs (90%), but with a relatively lower percentage (62%) in haV-CFs. A very small population of PDGFRβ⁺ cells which may represent nonmyocytes in the hPSC-CMs preparation (Fig. 6b). Immunolabeling of the cell preparations was performed to evaluate for other proteins associated with fibroblasts. The transcription factor GATA4 has previously been reported to be expressed in CFs but not in DFs[5], and we found the TE-7-labeled fibroblasts from hPSC-CFs, hfV-CFs, and haV-CFs were all GATA4 positive in contrast to hDFs which showed no immunolabeling for GATA4. hPSC-CMs were negative for the fibroblast marker TE-7, but positive for GATA4 (Fig. 6c). There were no cTnT⁺ cells detected in any of the fibroblast cell lines, but hPSC-CMs preparation showed abundant cTnT⁺ cells (Fig. 6c). Periostin is a matricellular protein associated with fibroblasts that is expressed in mouse CFs during embryonic development and downregulated in adult mouse CFs except following injury[42,46]. Furthermore, a recent study demonstrated that periostin is expressed in CFs differentiated from hPSC-derived epicardial cells[9]. Because periostin is a secreted protein, flow cytometry is problematic for detection of expressing cells, so we performed Q-PCR to examine *POSTN* expression. All of the fibroblast populations examined expressed *POSTN*, but hPSC-CM did not show expression (Fig. 6d). Immunolabeling for FSP1, another protein associated with fibroblasts, was detected in the majority of cells from all fibroblast cell lines and likewise, the ECM proteins collagen I and fibronectin were expressed across all

the fibroblast preparations (Supplementary Fig. 14). Thus, while hPSC-CFs share some properties with hDFs, they are most similar to hfV-CFs and haV-CFs in their morphology and marker expression, particularly in the pattern of expression of CD90 and GATA4.

**Gene expression of hPSC-CFs is most similar to primary CFs**. To characterize the gene-expression pattern in hPSC-CFs, we performed RNA-seq on hPSC-CFs and compared the total mRNA transcripts with those of haV-CFs, hDFs, hPSC-CMs, and hPSCs. Analysis of gene expression across all the samples showed the greatest similarity in the abundance of overall transcripts (19,084) between hPSC-CFs and primary haV-CFs calculated by Euclidean distance (Fig. 7a). Bioinfomatics analysis using the online tool AmiGO2 for cardiac-related genes (58) demonstrated cardiac factors were enriched in hPSC-CFs, haV-CFs, and hPSC-CMs with hPSC-CFs and haV-CFs showing the greatest similarity (Fig. 7b). For example, *BMP4*, *GATA4*, and *SOX17* were similarly expressed in hPSC-CFs and haV-CFs, but not expressed in hDFs. However, other key cardiac genes showed greater expression in hPSC-CFs compared to haV-CFs, including *HAND2*, *HEY1*, *ISL1*, and *NKX2-5* (Fig. 7b). We also analyzed the ECM-related gene expression in all samples given the role of ECM production in fibroblast biology. High expression of collagens (*COL1A1*, *COL1A2*, *COL3A1*, *COL5A1*, *COL5A2*, *COL6A1*, *COL6A2*, *COL6A3*, *COL8A1*, and *COL12A1*), fibronectin (*FN1*), the matrix metalloproteinases 1, 2, and 14 (*MMP1*, *MMP2*, and *MMP14*),

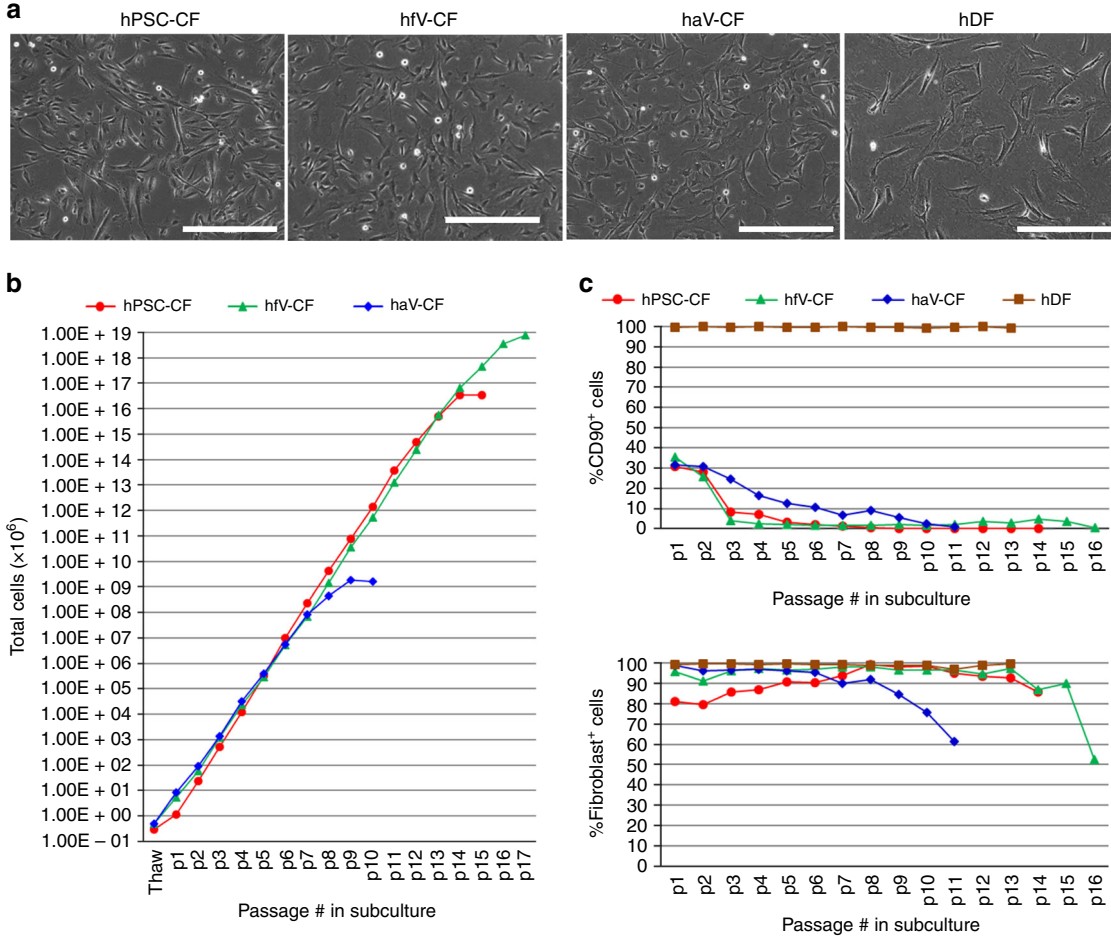

**Fig. 5** Comparison of CF and DF in morphology, proliferation and marker expression. **a** Phase contrast images of hPSC-CFs, hfV-CFs, haV-CFs, and hDFs. **b** Growth capacity of hPSC-CF, hfV-CF, and haV-CF during sequential passaging. **c** CD90 and the human fibroblast maker (clone TE-7) expression examined by flow cytometry in hPSC-CFs, hfV-CF, haV-CFs, and hDFs during passaging. Scale bars are 400 μm. The hPSC-CFs are from hiPSC line DF-19-9-11T. The primary hfV-CFs are from Cell Applications, Inc. The haV-CFs are from Lonza, NHCF-V. The hDF line is from skin biopsies from healthy donor

and tissue inhibitor of metalloproteinases 1, 2, and 3 (*TIMP1, TIMP2,* and *TIMP3*) were seen in all fibroblasts (Fig. 7c). However, some of the genes, including *COL4A1, COL4A2, COL4A5,* and *COL18A1* were more highly expressed in hPSC-CFs and haV-CFs than hDFs, and *MMP3* was expressed only in hDFs. Cluster analysis for the overall ECM-related gene expression again demonstrated hPSC-CFs and haV-CFs were the most closely related cell populations studied (Fig. 7c).

To compare the hPSC-CFs and haV-CFs in more detail, we performed gene ontology (GO) analysis using the online Functional Annotation Tool (DAVID Bioinformatics Resources 6.7, NIAID/ NIH) for the differentially expressed genes (Fig. 7d, Supplementary Data 1 and 2). Genes classified as extracellular region or extracellular region part are highly enriched in both hPSC-CFs and haV-CFs. Genes involved in membrane proteins are enriched in hPSC-CFs, whereas genes involved in cell–cell and cell–ECM attachment are enriched in haV-CFs. Interestingly, genes involved in pattern specification process, anterior/posterior pattern formation, and embryonic morphogenesis are enriched in hPSC-CFs consistent with the hPSC-CFs being more embryonic in phenotype (Fig. 7d).

To further examine the identity of hPSC-CFs, we compared our RNA-seq data with a recent publication which performed RNA-seq for gene-expression analysis of primary human fetal and adult CFs[47]. Unsupervised cluster analysis of ten representative genes differentially expressed in human fetal and adult

CFs identified by Jonsson et al.[47], demonstrated that hPSC-CF cluster most closely with hfV-CFs relative to the haV-CFs (Supplementary Fig. 15a). Furthermore, unsupervised cluster analysis of the cardiac factors (58 genes) showed the most similarity between the two haV-CFs, then grouped with the hfV-CFs and hPSC-CFs subsequently. The key cardiac factors expressed in hPSC-CFs that we identified above including *BMP4, GATA4, SOX17, HAND2, HEY1, ISL1,* and *NKX2-5* (Fig. 7b) are all expressed in hfV-CFs. Furthermore, *HEY1, ISL1,* and *NKX2-5* showed higher expression in hfV-CFs than in haV-CFs in the Jonsson et al. study[47], similar to the expression pattern in hPSC-CFs relative to haV-CFs in our data, consistent with a more embryonic phenotype of the hPSC-CFs (Supplementary Fig. 15b, Data 1). Interestingly, the transcription factor TBX20 has been identified in adult mouse and human CFs as a key marker[5], but TBX20 is expressed to significantly lower level in fetal human CFs[47]. Our data also showed TBX20 expression is significantly lower in hPSC-CFs (Supplementary Data 2), consistent with a more embryonic phenotype of the hPSC-CFs. ECM gene expression was similar across all the samples (Supplementary Fig. 15c). In summary, these gene-expression data demonstrate hPSC-CFs are most closely related to human native CFs relative to the other cell types studied, and the gene expression pattern is consistent with the hPSC-CFs having a more embryonic CF phenotype.

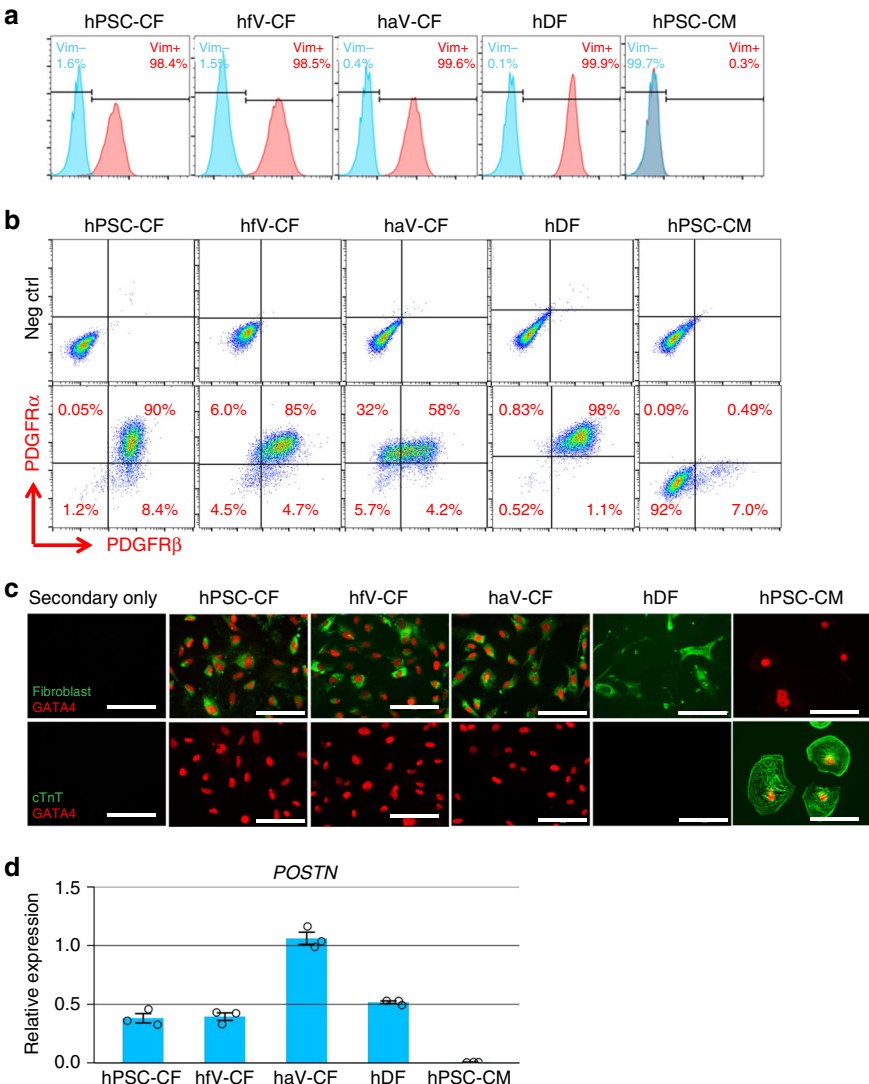

**Fig. 6** Representative markers expression in CF, DF and hPSC-CM. **a** Histogram plot of flow cytometry analysis for vimentin. **b** Flow cytometry analysis of co-labeling for PDGFRα and PDGFRβ. **c** Immunolabeling by antibodies for fibroblast (clone TE-7), cTnT and GATA4. **d** *POSTN* expression examined by qRT-PCR (*n* = 3 technical replicates). Data are mean ± SEM. Scale bars are 100 μm. All fibroblasts samples used in the measurement are from passages 3 to 7. The hPSCs are DF19-9-11T hiPSCs. The primary hfV-CFs are from Cell Applications, Inc. The haV-CFs are from Lonza, NHCF-V. The hDF line is from skin biopsies from healthy donor

**hPSC-CFs produce 3D ECM scaffold.** Generation of ECM is a major function of CFs. Therefore, we examined the ECM generated by the hPSC-CFs in comparison with the hfV-CFs, haV-CF, and hDFs following 10 days of high-density culture[48]. To examine the ECM production and assembly, the fibroblast cultures were fixed, but not permeabilized, and immunolabeled with the antibodies to collagen type I and fibronectin. The nuclei were stained with Hoechst. Confocal imaging demonstrated that all of the fibroblast preparations robustly produced collagen I and fibronectin (Fig. 8a). However, 3D reconstruction of the confocal images revealed that the hPSC-CFs, hfV-CFs, and haV-CFs produced multicell layer cultures while the hDFs produced only monolayer cultures (Fig. 8a, Supplementary Movies 1–4). The average depths of the multilayer hPSC-CF (~40 μm) and the hfV-CF (~44 μm) cultures were similar and greater than the haV-CF cultures (~24 μm) and the monolayer hDF cultures (~12.5 μm) (Fig. 8b). Although extracellular fibronectin and collagen I were evenly distributed throughout the monolayer hDF cultures, fibronectin was concentrated in the top layers of all of the CF cultures in contrast to the distribution of collagen I throughout the depth of the multilayer CF cultures (Fig. 8a). To confirm adequate antibody penetration and ECM protein labeling, multilayer hPSC-CF cultures were permeabilized before immunolabeling. The permeabilized hPSC-CF culture shows a distinct pattern of fibronectin immunolabeling with clear intracellular fibronectin detected as well as more fibrillar extracellular fibronectin concentrated in the upper layers (Supplementary Fig. 16, Movie 5). Polarization of extracellular fibronectin has been observed in other in vitro cultured cell systems and attributed to the binding of cell produced fibronectin with soluble fibronectin in the serum containing medium to generate insoluble fibronectin concentrated at the apical surface of the cultures[49–51]. Overall, these results demonstrate that high-density culture of the fibroblast preparations results in robust production of ECM proteins, but hDFs exhibit contact inhibition and limited proliferation forming a monolayer relative to hCFs that generate multilayer

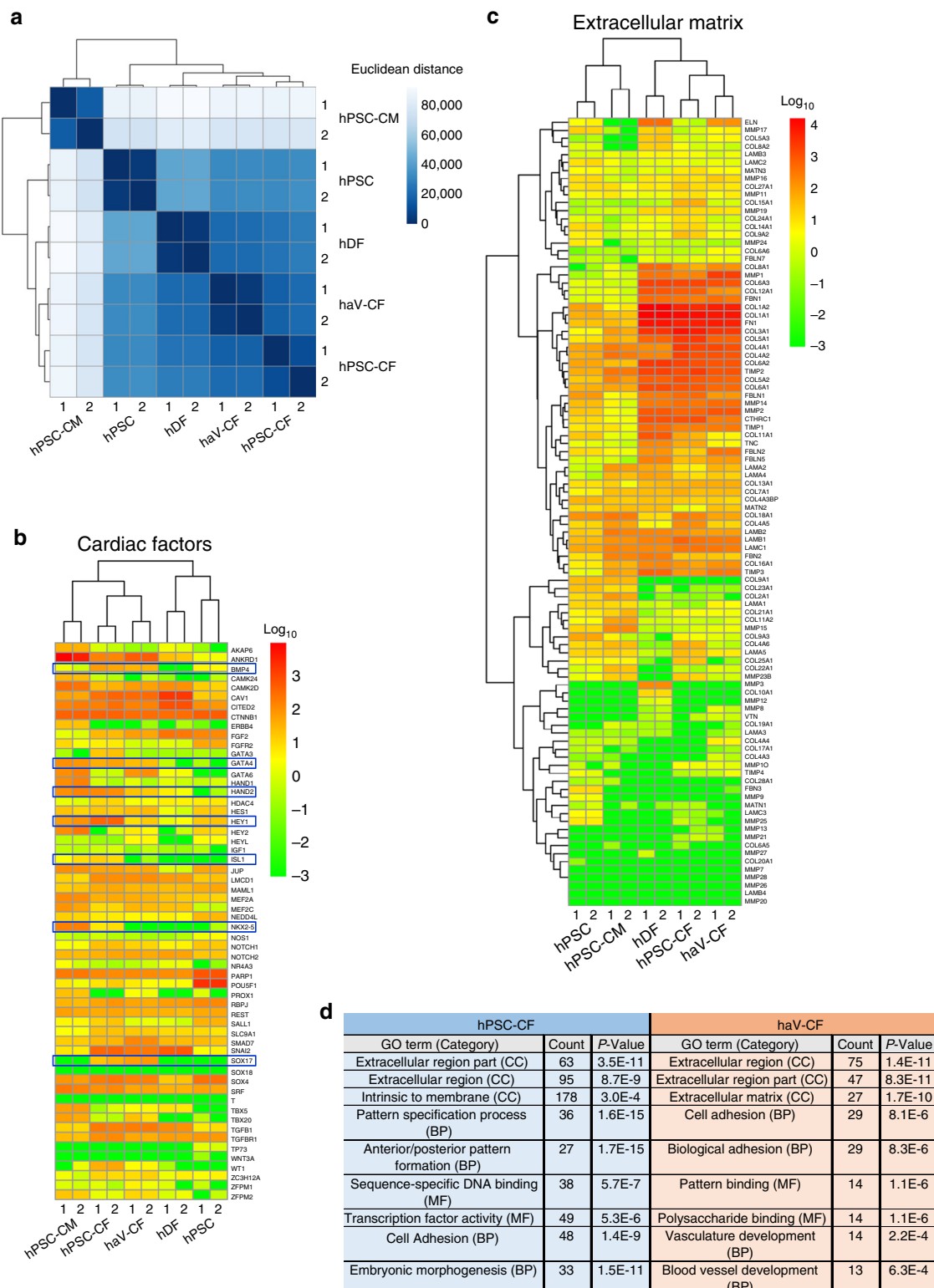

**Fig. 7** Gene expression by RNA-seq in CF compared to DF, hPSC-CM, and hPSC. **a** Dendrogram showing the similarity in the abundance of overall transcripts (19,084 genes) across samples of hPSC-CF, haV-CF, hDF, hPSC-CM, and hPSC that was calculated by Euclidean distance. **b** Heatmap of cardiac factors (58 genes) expression across samples of hPSC-CF, haV-CF, hDF, hPSC-CM, and hPSC organized by unsupervised cluster analysis. **c** Heatmap of extracellular matrix related gene expression (98 genes) across samples of hPSC-CF, haV-CF, hDF, hPSC-CM, and hPSC presented by unsupervised cluster analysis. **d** Gene ontology (GO) analysis using the functional annotation tool (DAVID Bioinformatics Resources 6.7, NIAID/NIH) for the differentially expressed genes in hPSC-CF and haV-CF (Supplementary Data 1 and 2) with calculated *p* values shown. The hPSC line used for RNA-seq are DF19-9-11T hiPSCs. The haV-CFs are from Lonza, NHCF-V, and hDFs are from 020a line. The TPM values from the RNA-seq were used for the analysis

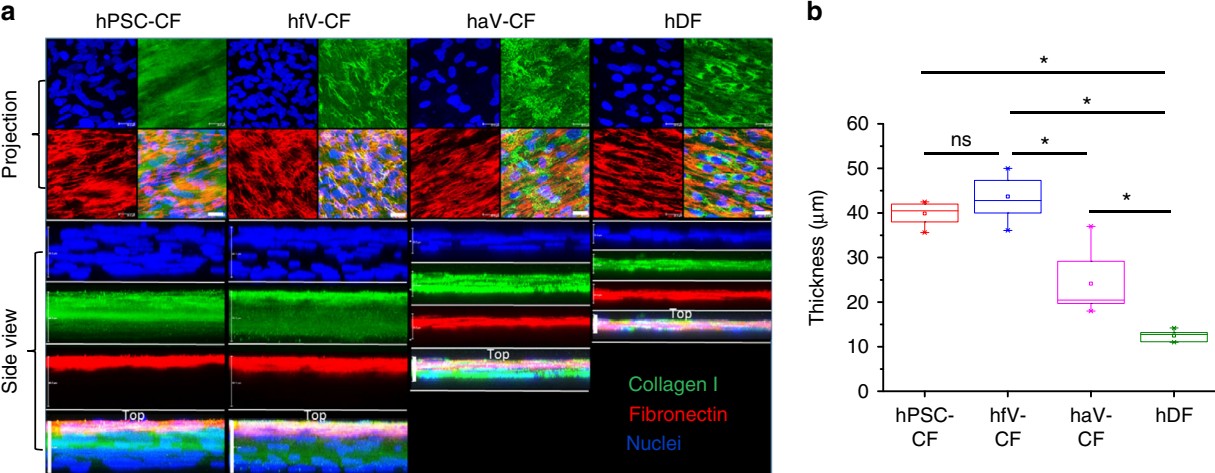

**Fig. 8** ECM production in CF and DF cultures. **a** Confocal imaging of high density cultures of hPSC-CF, hfV-CF, haV-CF, and hDF immunolabeling for extracellular collagen I and fibronectin. Z-scans are presented as projection images in the top panels and 3D reconstructions showing side views in lower panels. Scale bars in Projection are 25 μm. Scale bars in Side view of hPSC-CF and hfV-CF are 40 μm, haV-CF are 20 μm, and hDF are 15 μm. **b** Thickness of the 3D ECM scaffolds measured from 3D reconstructions of hPSC-CF ($n = 8$), hfV-CF ($n = 9$), haV-CF ($n = 12$), and hDF ($n = 6$) biological replicates. The box plots summarize the biological replicates with the box enclosing from first to third quartile, middle square indicating mean, line in box indicating median, and whiskers indicating outliers. *$P < 0.05$, one-way ANOVA with post hoc Bonferroni test

cultures. Furthermore, the thickness of the hPSC-CF cultures is most similar to hfV-CF cultures and significantly greater than the thickness of haV-CFs, which is consistent with the hPSC-CFs exhibiting a more embryonic CF phenotype.

**hPSC-CFs can transform to myofibroblasts.** In the setting of injury or disease, CFs can convert to myofibroblasts and contribute to remodeling of the heart. TGFβ1 signaling is a major stimulus for the transformation of CFs to myofibroblasts[52–56]. Angiotensin II (Ang II) is another agonist that stimulates myofibroblast conversion and fibrotic effects in CFs but typically not in DFs[57,58]. To evaluate for myofibroblast formation induced by both TGFβ1 and Ang II, we cultured the CFs in DMEM+10% FBS for 48 h in the presence or absence of TGFβ1 or Ang II. The hPSC-CFs, hfV-CFs, and haV-CFs grown with optimized fibroblast media exhibited low expression of α-smooth muscle actin (SMA^low) measured by both flow cytometry and immunocytochemistry. In contrast, hDFs in the confluent culture exhibited a SMA^high population by flow cytometry (Fig. 9, Supplementary Fig. 17). Totally, 48 h of culture in DMEM+10% FBS led to the emergence of a SMA^high population in all of the CF cultures. Furthermore, application of TGFβ1 increased the SMA^high population for the hPSC-CFs, hfV-CFs, and haV-CFs, and the SMA immunolabeling showed the strong SMA expressing stress fibers (Fig. 9). In contrast, the TGFβ1-treated hDFs showed the overall increase of SMA expression and a larger SMA^low population measured by flow cytometry, and the SMA immunolabeling also showed the increased SMA expression in the stress fibers (Fig. 9). The cell morphology of all the CFs from baseline conditions changed significantly from spindle-shaped, small cells to larger, more elongated cells (Fig. 9). Thus, all of the CF cultures exhibited comparable patterns of transformation to myofibroblasts based on the formation of a SMA^high population with prominent stress fibers in response to TGFβ1. The expression pattern of SMA also changed in hDFs in response to TGFβ1, but the pattern of response was distinct from CFs as measured by flow cytometry (Fig. 9). Ang II treatment induced the SMA^high population as observed with TGFβ1 treatment in both hPSC-CFs and hfV-CFs, but in hDFs Ang II resulted in a decrease in the

SMA^high population. Comparable results were observed in response to 100 and 500 nM of Ang II (Supplementary Fig. 18).

**Co-culture of hPSC-CFs effects hPSC-CM electrophysiology.** Given that CFs can influence the electrophysiological properties of the native heart, we examined the functional impact of the co-culture of hPSC-CFs, haV-CFs, or hDFs with hPSC-CMs monolayers. Various ratios of fibroblasts and hPSC-CMs were co-cultured, and representative images confirm different co-culture proportions using immunofluorescent labeling for cTnT (green) to identify hPSC-CMs and DAPI to label the nuclei of all cells (Fig. 10a). In functional experiments, co-cultured monolayers were labeled with a membrane potential sensitive dye (FluoVolt) to quantify the electrophysiological effects of co-cultured fibroblasts. Time–space plots of action potential propagation showed the differential impact of each fibroblast subtype on hPSC-CM monolayer function (Fig. 10b). Cultures of 100% hPSC-CMs showed regular spontaneous automaticity with uniform conduction across the cultures (Supplementary Movie 6). Comparison of 10 and 50% fibroblast co-culture demonstrated that the spontaneous rate was significantly faster for hPSC-CF co-culture compared to progressively slower rates for haV-CF and hDF co-cultures (Fig. 10b, c, Supplementary Movie 7a–c). Impulse conduction velocity was slowed by co-cultures with 10% of any of the fibroblast populations, although to a greater extent in the hDF co-cultures (Fig. 10b, c, Supplementary Movie 7a–c). Interestingly, at a ratio of 50% hPSC-CMs:50% fibroblasts, we observed fibrillatory conduction patterns in all of the monolayer co-cultures with hPSC-CFs (6/6) and in half of the monolayers with haV-CFs (2/4), while the same ratio of hDFs did not induce this arrhythmic pattern in any hPSC-CM monolayer co-cultures tested (0/6) (Fig. 10b, c, Supplementary Movie 8a–c). Given the different spontaneous rates observed for the various conditions, we used field stimulation to control rate and constructed action potential duration (APD) restitution curves for the monolayer co-cultures. The 100% hPSC-CM and 10% hPSC-CF co-cultures showed the shortest APD80s over the range of paced rates (Fig. 10d). These data demonstrated that the electrophysiological

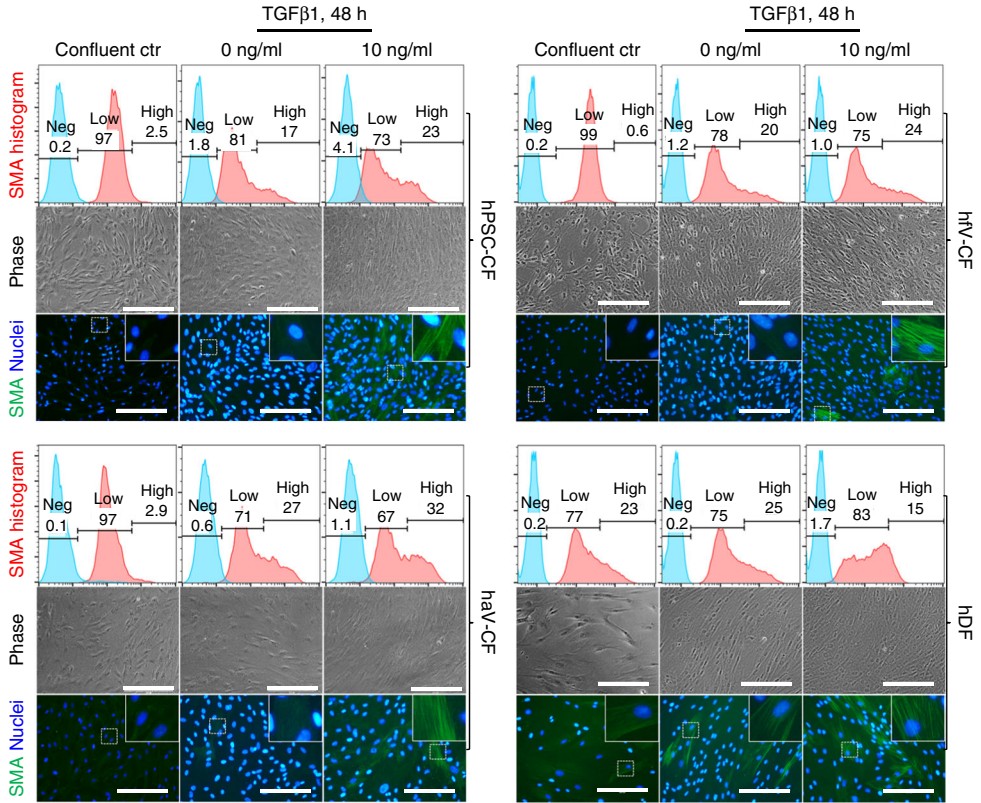

**Fig. 9** Myofibroblast transformation of CFs and DFs. Analysis of α-smooth muscle actin (SMA) expressing cells to assay for myofibroblast transformation from hPSC-CF, hfV-CF, haV-CF, and hDF in culture. All fibroblast cultures in passages 4–6 were treated with 10 ng/ml TGFβ1 for 48 h in DMEM+10% FBS medium. Expression of SMA was measured by both flow cytometry (upper panel) and immunolabeling (lower panel). Phase contrast images of representative cultures are shown in the middle panel. Phase contrast images and SMA immunofluorescence images are from the same biological samples, but not the same field of view. Scale bars are 200 μm. The hPSC-CFs are derived from DF19-9-11T hiPSCs. The hfV-CFs are from Cell Applications, Inc. The haV-CFs are from Lonza, NHCF-V, and hDFs are from 023a line

properties of the hPSC-CM monolayers are distinctly modulated by co-cultures with the different populations of fibroblasts.

## Discussion

In the present study, we demonstrate that stage-specific activation of Wnt and FGF signaling promotes the efficient differentiation of hPSCs via SHFPs to CFs. The hPSC-CFs share characteristics with native human CFs based on gene expression, cell morphology, proliferation, ECM production, myofibroblast transformation, and the distinct modulation of hPSC-CM electrophysiology. Although cell type-specific markers for CFs are not known, the combination of different markers present in the hPSC-CFs including expression of PDGFRα, vimentin, FSP1, and labeling with human fibroblast antibody along with GATA4 expression are consistent with a CF phenotype. Overall, we conclude that this monolayer-based protocol using defined conditions allows the scalable generation of SHF-derived hPSC-CFs.

In mammalian cardiac development, CFs arise from epicardium, endocardium, and neural crest progenitor populations[59]. These different fibroblast populations are concentrated in different regions of the heart and exhibit distinct contributions to cardiac development such as the role of endothelium-derived CFs in formation of the cardiac cushion and ultimately cardiac valves. The three embryological origins of CFs suggest that distinct fibroblast populations can be differentiated from hPSCs. The mesoderm-directed differentiation protocol argues against neural crest ectoderm derived CFs. Recent studies have described the differentiation of hPSCs to epicardial cells that can give rise to CFs[8–11]. These studies

provided evidence for generation of epicardial cells based on the expression of characteristic markers including *WT1*, *TBX18*, and *ALDH2*. In the present study, we found a different pattern of gene expression induced by FGF signaling in the developing progenitor populations with strong evidence for a SHFP population marked by persistent *ISL1* expression, high *HAND2*, and transient expression of *CXCR4*[43]. The SHFPs in development can give rise to endocardium via a TIE2 expressing angiogenic progenitor or a TIE2⁻ but ISL1⁺ population. The endocardium is a specialized endothelial population commonly distinguished by the expression of NFATc1[45]. Our studies also did not reveal clear expression of TIE2 or NFATc1 during intermediate stages, implying the differentiation of endocardial cells may require different signaling pathways than the FGF signaling in the GiFGF protocol. However, given the strong signal for SHF genes there is evidence for SHFPs and the expression of an endothelial-to-mesenchymal transition related gene in the GiFGF protocol, *SNAI2*, suggests the hPSC-CFs may have passed through a transient endocardial intermediate. Alternatively, the SHFPs may directly give rise to CFs as has been suggested for SHF-derived endocardial cells in which some endocardial cells arise via an endothelial/endocardial progenitor population and others arise directly from SHFPs[21]. Consistent with our findings of the SHF-derived hPSC-CFs, a recent study using genetic lineage tracing in mouse models demonstrated that SHFPs in vivo can give rise to CFs as can ESC-derived SHFPs in vitro[43].

The maturity of lineages derived from hPSCs remains a critical question in applications. For example, hPSC-CMs generated from existing protocols are relatively immature exhibiting an embryonic/fetal CM phenotype based on a number of key properties including

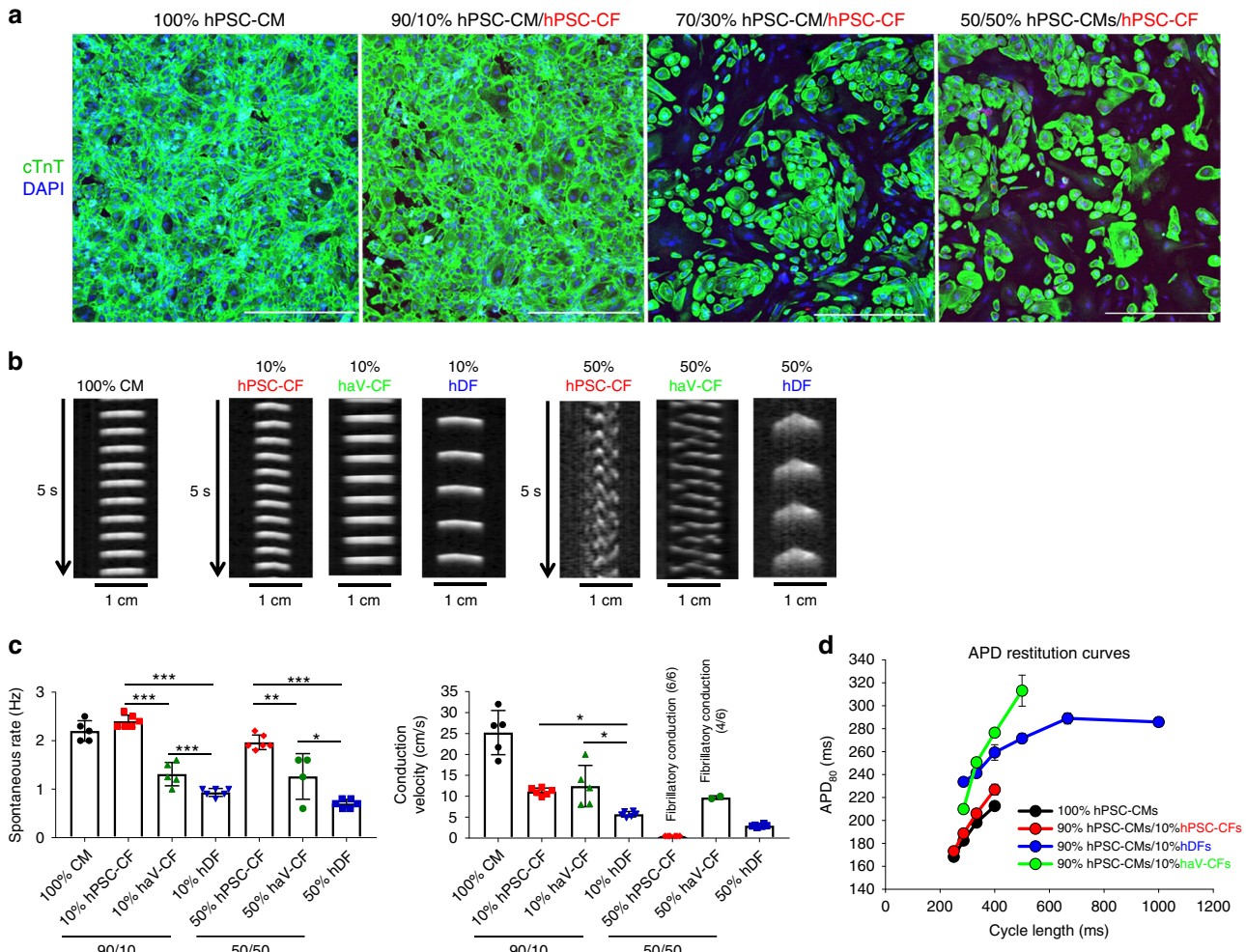

**Fig. 10** Effect of CFs or DFs on hPSC-CM electrophysiology in co-culture. **a** confocal images of different ratios hPSC-CMs and hPSC-CFs in culture together with immunolabeling for cTnT to identify hPSC-CMs and DAPI labeling for all cells present. Scale bars are 500 μm. **b** Time–space plots of optical mapping experiments. Postacquisition analysis of optical recordings was done by plotting the changes of fluorescence over time along a single line of each monolayer, similar to a confocal line scan. Differences in the spontaneous rates of activation are evident as are altered patterns of conduction with fibrillatory conduction observed with 50% hPSC-CF and haV-CF co-cultures with hPSC-CM. **c** Average spontaneous beating rate and action potential propagation velocity for 100% hPSC-CM monolayers ($n = 5$) compared to co-cultures of percentages and cell types indicated (red, hPSC-CF 10% ($n = 6$) and 50% ($n = 4$); green, haV-CF 10% ($n = 5$) and 50% ($n = 6$); blue, hDF 10% ($n = 6$) and 50% ($n = 6$)). Statistical comparisons were made between different fibroblast populations at each fixed mixture with ANOVA and Tukey's post hoc testing for significant differences as shown, ***$P < 0.0001$; **$P < 0.001$; and *$P < 0.01$. **d** Average action potential duration at 80% of repolarization (APD80) restitution curves generated by pacing indicated preparations over a range of different cycle lengths and measuring optical APD80. The hPSC-CF and hPSC-CMs are derived from H9-cTnT-GFP hESCs. The haV-CFs are from Lonza, NHCF-V, and hDFs are from skin biopsy from a healthy adult donor. Data are mean ± SEM with $n$ representing number of separate monolayers (biological replicates) tested

cell size and structure, myofilament composition, electrophysiology and metabolism[60–63]. In the case of hPSC-CFs, the issue of maturity is equally relevant. However, the CF-specific developmental milestones are not well defined. We observed the hfV-CFs and hPSC-CFs have a greater proliferative capacity than haV-CFs, a more uniform expression of PDGFRβ, and the ability to generate thicker 3D high density cultures. Furthermore, the gene expression pattern in the hPSC-CFs from our data is more similar to human fetal CFs relative to adult CFs from a recently published study[47]. Although these functional assays and gene expression studies are consistent with hPSC-CFs having a more embryonic phenotype, there are other possible explanations given the multiple sources of CFs that may exhibit differences in their relative abundance in fetal and adult human heart.

The ability to differentiate hPSCs to hPSC-CFs provides an unlimited supply of CFs for a range of research in cardiac biology and disease as well as potential clinical applications in tissue engineering and cardiac regeneration. Nevertheless, future studies will be needed to fully appreciate the properties of hPSC-CFs and how they relate to the heterogeneous and incompletely defined CFs present in the native heart.

## Methods
**Human PSC culture**. Human iPSC line DF19-9-11T and ESC lines H1 and H9-TnnT2-GFP were used in this study (WiCell Research Institute, cell line name: iPS DF19-9-11T; WA01; H9-hTnnT2-pGZ-D2, respectively). hPSCs were cultured on Matrigel (GFR, BD Biosciences) coated 6-well plates in either mTeSR1 (DF19-9-11T, H1) medium (Stem Cell Technologies) or E8 (H9-TnnT2-GFP) medium (ThermoFisher). Cells were passaged using Versene solution (Invitrogen) every 4–5 days. Cells were washed with DPBS without Ca$^{++}$/Mg$^{++}$ (Gibco) and incubated with 1 ml/well Versene solution at 37 °C for 3 mins. Aspirating Versene solution and adding 1 ml/well mTeSR1 or E8 medium to gently pipette up and down and scrape to cell clusters. Cell clusters

were pellet down by centrifugation at 1000 rpm for 3 mins, and resuspended in mTeSR1 or E8 medium, and plated in 6-well plate at the split ratio of 1:6 to 1:12. Mycoplasma tests were routinely performed (monthly) for all the cell culture in this study.

**Differentiation of hPSCs to CMs using GiWi protocol**. Human PSCs were dissociated with 1 ml/well Versene solution at 37 °C for 5 min, and seeded on Matrigel coated 6-well plates at the density of $1.5–2.0 \times 10^6$ cells/well in mTeSR1 medium supplemented with 10 μM ROCK inhibitor (Y-27632) (Tocris). Cells were cultured for 5–6 days in mTeSR1 medium until reaching 100% confluence when differentiation started (day 0). At day 0, the medium was changed to 2.5 ml RPMI+B27 without insulin (Gibco) and supplemented with 12 μM CHIR99021 (Tocris), and cells were treated in this medium for 24 h (day 1). After day 1, medium was changed to 3 ml RPMI+B27 without insulin and cells were cultured in this medium for 48 h (day 1-3). Seventy-two hours (day 3) after addition of CHIR99021, a combined medium was prepared by collecting 1.5 ml of medium from the wells and mixing with same volume of fresh RPMI+B27 without insulin medium and supplemented with 5 μM IWP2 (Tocris), and cells were treated with IWP2 in the medium for 48 h (day 3-5). The medium was changed to 3 ml RPMI+B27 without insulin on day 5. On day 7 the medium was changed to RPMI+B27 with insulin (Gibco), and the cells were fed every other day and cultured until day 20 for flow cytometry analysis.

**Differentiation of hPSCs to CFs**. Human PSCs were dissociated with 1 ml/well Versene solution (Gibco) at 37 °C for 5 min, and seeded on Matrigel (GFR, BD Biosciences) coated 6-well plates at the density of $1.5–2.0 \times 10^6$ cells/well in mTeSR1 medium supplemented with 10 μM ROCK inhibitor (Y-27632) (Tocris). Cells were cultured for 5–6 days in mTeSR1 medium with medium changes daily until they reached 100% confluence when differentiation started (day 0). At day 0, the medium was changed to 2.5 ml RPMI+B27 without insulin and supplemented with 12 μM CHIR99021 (Tocris) and cells were treated in this medium for 24 h (day 1). After day 1, the medium was changed to 2.5 ml RPMI+B27 without insulin (Gibco) and cells were cultured in this medium for 24 h (day 2). After day 2 but within 24 h (before day 3), the medium was changed to 2.5 ml of the CFBM medium (Supplementary Table 1) supplemented with 75 ng/ml bFGF (WiCell Research Institute). Cells were fed with CFBM+75 ng/ml bFGF every other day until day 20 when they were used for flow cytometry analysis and passaged.

**Culture of human CFs and DFs**. The hPSC-differentiated CFs (hPSC-CFs), primary human fetal ventricular cardiac fibroblasts (hfV-CFs, Cell Applications, Inc., cryopreserved HCF, fetal: 306-05f, lot 2666) and adult ventricular cardiac fibroblasts (haV-CFs, Lonza, NHCF-V, CC-2904, lot 0000401462) and primary dermal fibroblasts (hDFs, healthy donors, 020a and 023a lines, generated with informed consent as approved by UW-Madison Health Science IRB, protocol H-2008-0250) were used in this study. hPSC-CFs at 20 days of differentiation were dissociated with 0.05% Trypsin-EDTA (Gibco) and plated in 6-well noncoding plastic plates at the density of 300,000 cells/well in 2 ml of FibroGRO medium (Millipore EMD, SCMF001) plus 2% FBS (Gibco). The hPSC-CFs were fed every other day with 2 ml FibroGRO+2% FBS medium and passaged every 4–6 days using 0.05% Trypsin-EDTA. The plating density for passaging hPSC-CFs was 5800 cells/cm². The hfV-CFs were thawed at the plating density of 7000 cells/cm² in the Human CF Growth Medium (CFGM, Cell Applications, Inc., 316–500) and the haV-CFs were thawed at the plating density of 3500 cells/cm² in FGM-3 medium (Lonza, CC-4526) in noncoding plastic T75 flasks. The media were changed the next day after thaw and cells were fed every other day afterward. The hfV-CFs were cultured in both the CFGM and FibroGRO+2% FBS media and were passaged every 4–6 days using 0.05% Trypsin-EDTA. The haV-CFs were cultured with the FGM-3 medium and were passaged every 4–8 days using 0.05% Trypsin-EDTA. The subsequent plating density for passaging after thaw was 5800 and 3500 cells/cm² for hfV-CFs and haV-CFs, respectively. The primary hDFs were thawed in DMEM high glucose with GlutaMAX supplement (Gibco 10566-016) plus 10% FBS (Gibco 10437) in noncoding plastic T75 flasks at the density of 1900 cells/cm². The hDFs were cultured in the DMEM high glucose+10% FBS with medium change every other day and were passaged every 6–8 days using 0.05% Trypsin-EDTA. The subsequent plating density for passaging hDFs was 1900 cells/cm².

All fibroblasts were cryopreserved in 50% DMEM high glucose with GlutaMAX supplement (Gibco 10566-016), 40% FBS and 10% DMSO.

**Flow cytometry**. Cells were dissociated by incubation with 0.25% trypsin-EDTA (Gibco) plus 2% chick serum (Sigma) for 5 min at 37 °C. Cells were vortexed to disrupt the aggregates followed by neutralization by adding equal volume of EB20 medium[41]. Cells were counted and 0.5–1 million of cells were used for labeling. For labeling surface markers, cells were labeled either before, or after fixed in 1% paraformaldehyde at 37 °C water bath for 10 min in the dark. For labeling intracellular markers, cells were fixed in 1% paraformaldehyde and permeabilized in ice-cold 90% methanol for 30 min on ice. Cells were washed once in FACS buffer (DPBS without Ca++/Mg++, 0.5% bovine serum albumin (BSA), 0.1% NaN3), centrifuged and resuspended in about 50 μl FACS buffer with (for intracellular markers) or without 0.1% Triton X-100 (Sigma). For labeling surface markers,

appropriate amount of the conjugated antibodies according to the manufacturer's instruction were added to the cells in FACS buffer to make the final volume of 100 μl with FACS buffer, and incubated at room temperature in dark for 30 min. Cells were washed once with 3 ml FACS buffer and resuspended in 300–500 μl FACS buffer for analysis. For labeling intracellular markers, the primary antibody was diluted in 50 μl /sample FACS buffer plus 0.1% Triton X-100 and aliquoted to each sample for a total sample volume of 100 μl. Samples were incubated with the primary antibodies overnight at 4 °C. Please refer to Supplementary Table 2 for the primary antibodies used in this study. For the secondary antibody labeling, cells were washed once with 3 ml FACS buffer plus 0.1% Triton X-100 after the primary antibody labeling, centrifuged, and supernatant discarded leaving ~50 μl. Secondary antibody specific to the primary IgG isotype was diluted in FACS buffer plus 0.1% Triton X-100 in a final sample volume of 100 μl at 1:1000 dilution. Samples were incubated at room temperature in dark for 30 min, washed in FACS buffer and resuspended in 300–500 μl FACS buffer for analysis. Data were collected on a FACSCalibur (Beckton Dickinson) and Attune Nxt (ThermoFisher) flow cytometers and analyzed using FlowJo.

**Immunocytochemistry**. For imaging with EVOS microscope (Life Technologies), cells were cultured in either 6-well or 12-well plates. Cells were fixed with 4% paraformaldehyde for 15 min at room temperature, and permeabilized in 0.2% Triton X-100 (Sigma) for 0.5–1 h at room temperature. After fix and permeabilization, cells were blocked with 5% nonfat dry milk (Bio-Rad) in 0.2% Triton X-100 solution and incubated for 1.5–2 h at room temperature on a rotator followed by two washes with DPBS. Primary antibodies (Supplementary Table 2) were added in 1% BSA in DPBS solution with 0.1% Triton X-100, and incubated overnight at 4 °C. Cells were washed with 0.2% Tween 20 in DPBS twice and DPBS twice. Secondary antibody specific to the primary IgG isotype were diluted (1:1000) in the same solution as the primary antibodies and incubated at room temperature for 1 h in dark on a rotator. Cells were washed with 0.2% Tween 20 in DPBS twice and DPBS twice. Nuclei were labeled with Hoechst or DAPI.

**Quantitative RT-PCR**. Cell samples were collected using 0.25% trypsin–EDTA (Invitrogen) to remove the cells from cell culture plates. Total RNA was purified using QIAGEN RNeasy® Mini kit. Any genomic DNA contamination was removed by RNase-Free DNase Set (QIAGEN) with the RNeasy columns or by DNase I (Invitrogen) treatment for 15 min at room temperature. 500 ng of total RNA was used for Oligo (dT)20—primed reverse transcription using SuperScript™ III First-Strand Synthesis System (Invitrogen). Quantitative RT-PCR was performed using Taqman PCR Master Mix and Gene Expression Assays (Applied Biosystems, Supplementary Table 3) in triplicate for each sample and each gene. 0.5 μl of cDNA from RT reaction was added as template for each Q-PCR reaction. The expression of genes of interest was normalized to GAPDH.

**RNA-seq and data analysis**. Cell samples were collected using 0.25% trypsin–EDTA (Invitrogen) to remove the cells from cell culture plates. Total RNA was purified using QIAGEN RNeasy® Mini kit. Possible genomic DNA contamination was removed by RNase-Free DNase Set (QIAGEN) with the RNeasy columns. Duplicates for each sample were submitted for RNA-seq. Total RNA was qualified with the Life Technologies Qubit fluourometer (Q32857) and Agilent Bioanalyzer (G2940CA). One-hundred nanograms of total RNA was used to prepare indexed cDNA libraries with the Ligation Mediated Sequencing (LM-Seq) protocol[64]. Final indexed cDNA libraries were pooled with thirty uniquely indexed LM-Seq cDNA libraries per lane and sequenced on an Illumina HiSeq 2500 with a single 51 bp read and a 10 bp index read. Base-calling and demultiplexing were performed using Casava (v1.8.2). Sequences were filtered and trimmed to remove low quality reads, adapters, and other sequencing artifacts. The remaining reads were aligned to 19084 Refseq genes extracted from the Illumina iGenomes reference, selecting only those with "NM_" annotations. Bowtie (v 0.12.9) was used for alignment, allowing two mismatches in a 28 bp seed[65]. Reads with more than 200 alignments were excluded from further analysis. RSEM (v1.2.3) was used to estimate isoform and relative gene expression levels (transcripts per million or "TPM")[66]. Genes for specific functional groups were selected by searching online tool AmiGO2 (http://amigo.geneontology.org) using key words and manually confirmed. Heatmaps were generated using pheatmap_1.0.8 in R version 3.2.3. The similarity heatmap showed the Euclidean distances between samples which were calculated using R dist function on normalized read counts.

**ECM production and confocal imaging**. Confluent hPSC-CFs, primary hfV-CFs and haV-CFs, and hDFs were dissociated with TrypLE Express (Gibco) and seeded in μ-Dish 35 mm ibidi dish (Cat# 81156) at the density of $1.25 \times 10^6$ cells/dish. Cells were grown for 10 days with medium change every other day. hPSC-CFs and primary hfV-CFs were grown with FibroGRO medium (Millipore EMD, SCMF001) plus 2% FBS Gibco, primary haV-CFs were grown in FGM-3 media (Lonza), and hDFs were grown in DMEM high glucose with GlutaMAX supplement (Gibco) plus 10% FBS (Gibco) medium. On day 10, cells were fixed with 4% paraformaldehyde for 15 min at room temperature, and stored in DPBS at 4 °C. The ECM proteins including collagen and fibronectin were co-labeled with anti-COL1A (COL-1) (Santa Cruz Biotechnology, sc-59772) and anti-Fibronectin

(A-11) (Santa Cruz Biotechnology, sc-271089) antibodies without permeabilization of the cells to examine the ECM using the immunolabeling protocol in the 'Immunocytochemistry' section. The cellular fibronectin and SMA expression were examined in the permeabilized cells and the antibodies were co-labeled using the same immunolabeling protocol in the "Immunocytochemistry" section. Cells were imaged using the confocal microscope (Leica TCS SP5 system).

**Co-cultures of hPSC-CMs and hPSC-CFs, haV-CFs or hDFs.** hPSC-CMs were cultured with hPSC-CFs, haV-CFs, or hDFs at varying percentages: 100, 90, 70, 50, or 0%. These co-cultures were plated at a density of 250,000 cells per 200 μl drop of EB20 medium (80% DMEM/F12, 0.1 mmol/L nonessential amino acids, 1 mmol/L L-glutamine, 0.1 mmol/L β-mercaptoethanol, 20% FBS and 10 μmol/L blebbistatin) on 18 × 18 mm pieces of polydimethylsiloxane silicone sheeting (SMI; Saginaw, MI) coated with Matrigel (500 μg/mL, Corning) and grown for 3 days kept at 37 °C in 5% $CO_2$[67]. At this point the media was changed to RPMI (Gibco) supplemented with B27 (Gibco) and replaced every 2 days for 4 days prior to experimentation.

**Optical mapping.** Optical action potentials and time–space plots were recorded using FluoVolt™ membrane potential probe (F10488, Life Technologies)[67–70]. After a 30 minute incubation period, monolayers were transferred to Hank's balanced salt solution (HBSS with calcium and magnesium, Gibco) for optical mapping recording and kept at 37°C for the duration of the experiment. All monolayers containing hPSC-CMs displayed spontaneous pacemaker activity which was recorded using a CCD camera (Red-Shirt Little Joe, Scimeasure, Decatur, GA, 80 × 80 pixels) with the appropriate emission filters and LED illumination[68]. A total of 10 s movies were taken at 200 fps (LabWindows Acquisition, National Instruments, Austin, TX, USA) which were amplified, filtered and digitized for offline analysis. Briefly, time sequence plot graphs of fluorescence intensity over time were generated from average action potentials and used for calculation of action potential duration after signal filtering. In addition, optical mapping movies were processed with Scroll to generate activation maps and calculation of conduction velocity after providing spatial calibration to the software. Repetitive stimuli (duration, 5 ms; strength, twice diastolic threshold) was applied by field stimulation using silver electrodes. Monolayers were paced faster than the spontaneous beating rate, which was different depending on the co-culture with hPSC-CFs, haV-CFs, or hDFs (Fig. 10c, d). Stimulation frequency ranged from 1000 to 285.7 ms cycle length.

**Statistics.** Sample size varies and was determined based on anticipated scatter of the data, but formal power calculations were not done given the lack of previous comparable datasets from which to estimate variance. Data are presented as mean ± standard error of the mean. Technical and biological replicates are as indicated for each dataset. For datasets with normal distributions, statistical significance was determined by Student's $t$ test for two groups or one-way ANOVA for multiple groups with post hoc test using Bonferroni and Tukey methods. Statistical analysis was performed using Origin, v9, $P < 0.05$ was considered statistically significant.

**Reporting summary.** Further information on research design is available in the Nature Research Reporting Summary linked to this article.

## Data availability
The RNA-seq data for this study are deposited in GEO with the accession code GSE126260. All other relevant data generated in this manuscript that support the findings of this study are available upon request from the authors.

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

## Acknowledgements

We thank Jennifer Bolin and Scott Swanson for the help of the RNA sequencing, Pei-cheng Jing for assistance of the statistical analysis of the RNA-seq data, Guadalupe Guerrero-Serna for immunostaining of co-culture monolayers, Pratik Lalit's discussion and input, and Jayne Squirrell and Kevin Eliceiri for consultation on multiphoton imaging. We acknowledge the Thomson lab RNA sequencing team, Flow Cytometry Core at UW-Madison provided support. J.L.C. received funding from Coordenação de Aperfeiçoamento de Pessoal de Nível Superior and Fundação de Amparo à Pesquisa do Distrito Federal. The work was funded by NIH R01 HL129798 (T.J.K.); NIH U01 HL134764 (T.J.K.); S10RR025644 (T.J.K.); and the UW Institute for Clinical and Translational Research, grant UL1TR000427, from the Clinical and Translational Science Award of the NCATS/NIH.

## Author contributions

T.J.K and J.Z. conceived the study and wrote the paper. J.Z. designed and carried out the experiments, and analyzed the data. R.T., G.C.K. and J.L.C. did most of the cell culture and differentiation and the PCR. K.F.C., A.M.D., T.J.H. and J.J. designed, performed the co-culture work and analyzed the data. E.C.R., E.G.S. and A.N.R. performed the ECM production. J.A.T. contributed to conception, writing, and funding of the project.

## Additional information

**Competing interests:** T.J.K. is a consultant for Cellular Dynamics International. E.G.S. and A.N.R. are co-founders and have a financial interest in Cellular Logistics Inc. The remaining authors declare no competing interests.

