## [Peer Review File · Nature Communications]

Reviewers' comments:

Reviewer #1 (Remarks to the Author):

The authors have optimized a protocol for differentiation of human cardiac fibroblasts (CFs) from human pluripotent stem cells (hPSCs) through generation of second heart field progenitors, mediated by Wnt and FGF signaling. They acknowledge that most of the fibroblasts are epicardial in origin with a minor contribution from endocardium and neural crest lineages and they cite a paper in which efficient differentiation of epicardial cells from hPSCs was obtained (Witty, A.D. et al 2013), although with a poor rate of differentiation to CFs. They hypothesize that, given the mesodermal origin of all the cardiac lineages, induction of mesodermal committed progenitors from hPSC may lead to more efficient generation of CFs than through epicardial differentiation.

The authors do not comment on the relevance of producing fibroblasts in vitro, which are easily obtainable from human heart biopsies or disposed heart material from cannulations and other procedures. They also do not discuss the previous report by Iyer et al (Development, 2015) documenting robust derivation of epicardium and its differentiated smooth muscle cell progeny from human pluripotent stem cells by differentiating hPSCs to lateral plate mesoderm and subsequently epicardial cells, which can be efficiently differentiated in either smooth muscle cells or fibroblasts (over 80% of periostin+ cells). While this detracts from the premise of the study, it would be interesting to compare the CFs obtained with the two protocols. As cardiac fibroblasts have already been produced before the report lacks novelty, although they have optimized a protocol to obtain fibroblasts, providing technical advances and their observation that cardiac fibroblasts differentially modulates CM electrical activity is intriguing.

Specific comments:

1. The authors adopt a published protocol for CM differentiation (GiWi protocol) and Figures 1a,b and suppl 1 are FACS plots to show changes in the expression of mesodermal markers in the first 5 days of differentiation. They define cardiac cardiac mesodermal progenitors as the Brachyurydown/CD90low stage, based on higher % of cells positive for ApelinR and PDGFRa/KDR (FACS analysis), but their first "justification" is in line 75: "showed upregulation of MESP1 mRNA expression (Fig. 2b) which is expressed in cardiac mesodermal progenitors." However, qPCR results in Fig. 2b refer to different stages, d6-d20, of a different protocol (GiFGF). This is confusing and needs clarification. Regarding the FACS plots: what is the difference between "cells only" and "d0"? Fig 1c-e shows the experimental design of their differentiation protocol, and changes in cardiac (MF20) and fibroblast (anti-human fibroblasts antibody) markers by FACS in response to different doses of bFGF, starting at different time points. I, II, III, should be treatment starting from d2, d3, d4 respectively, but it's not well explained in the figure nor in the legend. Moreover, the percentage of fibroblasts seems less than what shown in the Iyer et al 2015 paper mentioned above.

2. In Fig 2 they profile cells at different stages of their new differentiation protocol, GiFGF, by qPCR. It would have been good to have human cardiac fibroblasts as a positive control for the qPCR here, but they also provide RNAseq data (Fig.4). If they want to make the claim that they are generating mostly progenitors of the second heart field, with minimal expression of first heart field and epicardial markers, Suppl Fig2 is quite important, and that panel might be reorganized and merged with Fig 2b.

3. In Fig 3 they start comparing their CFs with primary human cardiac fibroblasts and dermal fibroblasts, they change acronym for their cells to hPSC-CF, and they apply their differentiation protocol to 2 ES and to iPS cell lines. They claim that hPSC-CF are more similar to primary hCF than to hDF based on FACS and ICC. Growth curves are presented comparing primary hCF (now called NHCF-

V) with one line of hPSC-CF, and they conclude that hPSC-CF also become senescent and stop growing after 10 passages: the two curves look quite different and the one for hPSC-CF is still pretty straight, it doesn't seem they have reached the plateau yet so their explanation is not convincing. Fig 3e is slightly random/out of place: it's no longer a comparison with adult fibroblasts but a comparison of the two protocols, GiWi and GiFGF (d20 and after 5 passages), to show again increased expression of the fibroblast marker, almost absent expression of MF20 in the GiFGF protocol and very low level of endothelial cells from both protocols (CD31 for some reason is plotted differently: histogram instead of pseudo-color plot). This could be integrated into the protocol section.

4. Fig 4 compares different type of fibroblasts via RNAseq and the 3D ECM scaffold produced by the. The message again is that hPSC-CF and NHCF-V cluster together compared to hDF, but no comment is made of the differences among them. For example, the cultures of hPSC-CF are thicker, with more cellular layers than the adult ones (4 versus 2 from the images), this could be another indication of "immature" phenotype, maybe the cells are less susceptible to contact inhibition or just more proliferative, but these differences should be acknowledged. Also, from the heat map it looks like hPSC-CF express more *Nkx2-5*, *WT1*, *Isl1*, *Hey1*, *Hand2* and less *HAND6* compared to the adult ones, suggesting a more "immature" phenotype? It is surprising that their differentiated CFs do express *WT1* suggesting an epicardial phenotype, and that there is no signal for *Tbx20* which was an abundant CF transcript in a previous report (Furtado et al, *Circ Res*, 2014). *Col1* and fibronectin expression do not seem to overlap with the cell layer for hPSC-CFs, but do for NHCF-V and hDFs, which is unexplained. Note that *Gata4* is NOT specific to cardiac fibroblasts; it is expressed by other organ fibroblasts and this should be corrected.

5. The data in Fig 5 are very interesting: they measure differences in electrical impulse propagation in mixed cultures of hPSC-CMs with either hPSC-CFs or hDF. For some reason, adult cardiac fibroblasts disappear from this assay, which would have been a good control. The supplementary movies should be relabeled as S1a, S1b, S2a, S2b, it's not easy to understand what is what.

Minor comment:

Apart from a clear inconsistency with the use of acronyms, typos and unclear figure legends, the guidelines for the journal were not followed; for example there shouldn't be references in the abstract etc.

Reviewer #2 (Remarks to the Author):

Zhang et al developed the protocol to induce CFs from human PSCs by directed differentiation. Sequential modulation of Wnt and FGF signaling induced second heart field progenitors that efficiently differentiated into hPSC-CFs. However, the cells did not express epicardial genes before differentiation into CFs. They also showed that the hPSC-CFs were similar to native heart CFs than dermal fibroblasts in gene expression and matrix production. Co-culture of hPSC-CFs with CMs demonstrated that hPSC-CFs were different from hDFs in modulation of hPSC-CM electrophysiology. Although the concept is interesting, it remains unclear why the hPSC-CFs did not express epicardial genes before CF differentiation. Moreover, if the authors can show difference between diseased-CFs and healthy-CFs from PSCs, the hPSC-CFs may be useful for disease modeling and drug discovery.

Major Points:

1. Fig 2.

Why the epicardial genes were not upregulated during differentiation? Given that a large population of

CFs are derived from the epicardium, it is important to determine the path of CFs in this protocol.

2. Fig 4, 5

The hPSC-CFs and hCFs resulted in polarized and thick matrix production in contrast to hDFs. What is the mechanism of this difference? Moreover, it would be important to determine the molecular mechanism for different electrophysiology in co-cultured hPSC-CMs between hPSC-CFs and hDFs.

3. Fig 5

Fibroblasts change the property by sequential passages and culture conditions. It remains unclear whether the hPSC-CFs are truly useful for disease modeling and drug discovery. Demonstrating the difference between diseased-CFs and healthy-CFs from human PSCs may be informative.

Reviewer #3 (Remarks to the Author):

The central objective of this manuscript was to develop an efficient method for obtaining cardiac fibroblasts from human PSCs by manipulating Wnt and FGF signaling. The authors strategy was to form cardiac mesodermal progenitors through routine inhibition of Wnt signaling as done for inducing myocyte differentiation and tested whether FGF stimulation could bias the differentiation of these progenitors to non-myocyte cell types. After examination of dose-response curves with recombinant FGF treatment the authors had made spindle shaped cells that secrete extracellular matrix and express markers associated with cardiac fibroblasts like Vimentin, Periostin, FSP1 and a little TCF21. Co-cultures with the various fibroblasts and myocytes demonstrated that the type and quantity of fibroblasts in co-culture remodel the myocytes action potential. The impact of the paper is potentially high as the ability to have a human model of fibroblast-based disease, as there is a growing need to find ways of treating cardiac fibrosis. The technology is also novel and could help pioneer new developments in the field, so this paper is of tremendous value. The manuscript does fall short in proving that the GiFGF protocol is really making cardiac fibroblasts, and what features of cardiac fibroblasts the cells are truly modeling.

The biggest problem with the paper, which is a problem for the field in general, is the genetic heterogeneity of fibroblasts within the heart and what genetic signatures correlate with given fibroblast functions. The authors make a decent attempt at trying to comparing the GiFGF fibroblasts to the Lonza human cardiac fibroblast cell line, but this particular line is a mixed population of valve, ventricular, and atrial fibroblasts which we know at least in the mouse have very different genetic signatures and functions, making it challenging to know which fibroblast population the GiFGF fibroblasts actually models. Moreover, the genetic signatures of these Day 20 fibroblasts have some overlap with well characterized mouse ventricle populations (ie. expression of vimentin, periostin, a little TCF21) yet they don't completely match cardiac fibroblast signatures either. In addition, the co-culture experiments while interesting and important are too early and don't fit the papers context. It would be better to spend that experimental space proving the GiFGF fibroblasts truly function like cardiac fibroblasts by testing whether they respond similarly to drugs like TGFb or Angiotensin II, or determining what state of maturation these fibroblasts are in. Finally, the authors claim efficient induction of the cardiac fibroblast phenotype which is based on flow sorting with a fibroblast surface marker TE-7; however, the specificity of TE-7 is questionable as many fibroblast surface markers also bind to dendritic cells (CD11b) or other hematopoietic cells. Validation of the purity of the cell population labeled by TE-7 would be helpful towards understanding the efficiency of GiFGF-mediated differentiation. Below are additional comments that also complicate the interpretation of this cardiac fibroblast differentiation protocol.

- In combining the expression data for the GiFGF induced fibroblast differentiation these cells seemed to be labeled by a little TCF21, Vimentin, Periostin, and FSP1 gene expression. Recent lineage tracing experiments have demonstrated that in adult cardiac fibroblasts Periostin only turns on with injury and that FSP1 actually labels hematopoietic cells. Moreover, Periostin gene expression has been reported in non-fibroblast cells when they are grown in culture. This is of concern given that traditional epicardial lineage marks aren't expressed nor were data provided to suggest they are more like endothelial derived cardiac fibroblasts that have Tie2 or NfactC1 expression. As stated these cells look like they recapitulate the second heart field development but whether fibroblasts in that region have different genetic signatures and functions is not well understood. So again its unclear what this GiFGF population is modeling. Can the authors identify additional markers like PDGFR α or Hey1 that might add support for these cells being cardiac fibroblasts? Or, the authors could compare these cells to fibroblasts obtained from the second heart field.
- It's clear these cells are making matrix but that can happen with many cell populations once they get into culture. Can the authors provide additional functional data that could prove these cells behave like cardiac fibroblasts? As stated, above perhaps looking at function with respect to their response to mechanical or chemical stimuli could answer this question.
- The authors need to add other fibroblasts populations as a basis for comparison given that the Lonza cell and dermal cells while human don't give enough specificity to the ventricle. Ideally the authors could get some human ventricular fibroblasts but even comparisons to neonatal and adult mouse ventricular cardiac fibroblasts would help with anatomical specificity and to define the fibroblasts stage of maturation. This may also help sort out why these fibroblasts are expressing periostin as this is more of the profile for neonates or injured hearts.
- The authors should validate the TE-7 antibody and show that it only labels cardiac fibroblasts.

Minor comments

- An important consideration in examining markers and fibroblast function are whether these fibroblasts convert to myofibroblasts with classic triggers and why these fibroblasts aren't activating during the 20 days of culturing on plastic.

The authors have optimized a protocol for differentiation of human cardiac fibroblasts (CFs) from human pluripotent stem cells (hPSCs) through generation of second heart field progenitors, mediated by Wnt and FGF signaling. They acknowledge that most of the fibroblasts are epicardial in origin with a minor contribution from endocardium and neural crest lineages and they cite a paper in which efficient differentiation of epicardial cells from hPSCs was obtained (Witty, A.D. et al 2013), although with a poor rate of differentiation to CFs. They hypothesize that, given the mesodermal origin of all the cardiac lineages, induction of mesodermal committed progenitors from hPSC may lead to more efficient generation of CFs than through epicardial differentiation.

- The authors do not comment on the relevance of producing fibroblasts in vitro, which are easily obtainable from human heart biopsies or disposed heart material from cannulations and other procedures.

Thank you for recognizing this important oversight in explaining the rationale of the study. In the Introduction, we now state, “Given the role of CFs in cardiac biology and disease, investigators have isolated and studied primary CFs from surplus patient material obtained at the time of cardiac surgery, cardiac biopsy, or autopsy. However, these primary CFs undergo senescence with sequential passaging and so limited numbers of genetically identical primary CFs are available for study or for tissue engineering applications that require large numbers highly qualified CFs. In the case of many inherited forms of heart disease, primary CFs are not readily available because patients typically do not undergo cardiac surgical interventions. Thus a robust, reproducible, and even patient-specific source of genetically identical human CFs is desirable.”

- They also do not discuss the previous report by Iyer et al (Development, 2015) documenting robust derivation of epicardium and its differentiated smooth muscle cell progeny from human pluripotent stem cells by differentiating hPSCs to lateral plate mesoderm and subsequently epicardial cells, which can be efficiently differentiated in either smooth muscle cells or fibroblasts (over 80% of periostin⁺ cells). While this detracts from the premise of the study, it would be interesting to compare the CFs obtained with the two protocols.

Thank you for noting this oversight, and now we directly address this publication. Consistent with our report, Iyer et al found FGF2 signaling alone failed to promote epicardial gene expression from lateral plate mesoderm in their effort to differentiate hPSCs to epicardial cells. This was further supported by a recent report from Weissman group (Loh et al., Cell 2016) showing that FGF2 signaling inhibits formation of (pro)epicardial cells when applied at the cardiac mesoderm stage in differentiation of hESCs. Iyer et al. demonstrated that their protocol could differentiate hPSCs to fibroblasts with the periostin⁺ cells reaching to 80% of the cells present. Iyer et al., provided no further characterization of the fibroblast population. In our revised manuscript, we also demonstrate that periostin is expressed by our hPSC-CF population based on mRNA analysis (Fig. 5f) because flow cytometry for this secreted protein is difficult to interpret. We agree that comparing epicardial-derived CFs as in Iyer et al., with our SHF hPSC-CF is an interesting experiment, but it is beyond the scope of this initial paper defining the differentiation protocol and initial characterization of these hPSC-CFs.

- As cardiac fibroblasts have already been produced before the report lacks novelty, although they have optimized a protocol to obtain fibroblasts, providing technical advances and their

observation that cardiac fibroblasts differentially modulates CM electrical activity is intriguing.

Yes, differentiation of cardiac fibroblasts from hPSC-derived epicardial cells has been described previously (Witty et al. Nat Biotech, 2014; Iyer et al., Development 2015; Bao et al., Nat Biomed Eng, 2016; Gaudin et al., Stem Cell Reports, 2017). However, we hope the reviewer will appreciate that we describe a distinct pathway to differentiate hPSCs to CFs via second heart field progenitors, and thus produce CFs that have a distinct developmental origin from those previously described. Furthermore, we generate CF with demonstrated higher purity and yield than previously previous protocols. Moreover, none of the previous studies provided detailed characterization or any functional analysis of the epicardial-derived hPSC-CFs as our current report does, particularly the distinct modulation of the electrophysiology of hPSC-CMs by hPSC-CFs. Thus, we suggest that there is significant novelty in the presented study including the developmental lineage producing CFs, the highly efficient protocol, and the detailed functional characterization of the CFs.

Specific comments:

1. The authors adopt a published protocol for CM differentiation (GiWi protocol) and Figures 1a,b and suppl 1 are FACS plots to show changes in the expression of mesodermal markers in the first 5 days of differentiation. They define cardiac cardiac mesodermal progenitors as the Brachyurydown/CD90low stage, based on higher % of cells positive for ApelinR and PDGFra/KDR (FACS analysis), but their first “justification” is in line 75: “showed upregulation of MESP1 mRNA expression (Fig. 2b) which is expressed in cardiac mesodermal progenitors.” However, qPCR results in Fig. 2b refer to different stages, d6-d20, of a different protocol (GiFGF). This is confusing and needs clarification.

Thank you for highlighting this confusion in our presentation of the data. We have revised Figure 1 to include the time course of gene expression from the GiWi protocol in Fig. 1c, which is now consistent with the text on the manuscript describing the increase in MESP1 mRNA expression indicating the generation of cardiac mesodermal progenitor cells.

Regarding the FACS plots: what is the difference between “cells only” and “d0”?

Day 0 cells refer to the day of the protocol described in the schematic in the new Figure 1a. It is simply the start of the protocol. Subsequent days of the protocol are likewise labeled. In contrast, ‘cells only’ refers to our controls for the flow cytometry, which were either without primary antibody or with an isotype control as appropriate. To clarify these as controls, we have changed the label to ‘Neg ctrl’ for negative control. We clarify this in Figure 1 legend by stating, “No primary antibody controls or isotype controls were performed for each time point, and the day 0, no primary antibody control (Neg ctrl) is shown as an example.”

Fig 1c-e shows the experimental design of their differentiation protocol, and changes in cardiac (MF20) and fibroblast (anti-human fibroblasts antibody) markers by FACS in response to different doses of bFGF, starting at different time points. I, II, III, should be treatment starting from d2, d3, d4 respectively, but it’s not well explained in the figure nor in the legend. Moreover, the percentage of fibroblasts seems less than what shown in the Iyer et al 2015 paper mentioned above.

We have clarified the labeling of these time points as stated in the new Figure 2 legend where we now present these data stating: “a. Schematic for testing the concentration-dependent effect of bFGF addition to stage-specific progenitors generated by the GiWi protocol beginning bFGF on day 2, 3, or 4, corresponding to labeled protocols I, II and III. Gray lines indicate RPMI medium + B27 supplement; blue lines indicate cardiac fibroblast basal medium (CFBM); green lines indicate CHIR treatment; red lines indicate bFGF treatment; orange line indicates IWP treatment.”

The average percentage of fibroblasts showed in Fig. 1c from the iPSC line 19-9-11 was 75%±4% which is comparable to the percentage of 81.2% POSTN⁺ cells from the single flow cytometry experiment presented by Iyer et al (Fig. 6B). Thus, as best as we can compare, the purity of CFs is similar.

2. In Fig 2 they profile cells at different stages of their new differentiation protocol, GiFGF, by qPCR. It would have been good to have human cardiac fibroblasts as a positive control for the qPCR here, but they also provide RNAseq data (Fig.4). If they want to make the claim that they are generating mostly progenitors of the second heart field, with minimal expression of first heart field and epicardial markers, Suppl Fig2 is quite important, and that panel might be reorganized and merged with Fig 2b.

Thank you for this excellent suggestion. We have merged the Fig. 2 and Suppl Fig. 2 into the new Figure 3 panels a. and b. We have revised the text accordingly.

3. In Fig 3 they start comparing their CFs with primary human cardiac fibroblasts and dermal fibroblasts, they change acronym for their cells to hPSC-CF, and they apply their differentiation protocol to 2 ES and to iPSC cell lines. They claim that hPSC-CF are more similar to primary hCF than to hDF based on FACS and ICC.

We apologize about the confusion with the acronyms. We are consistent now with the hPSC-CF acronym, and we also more clearly label the primary cardiac ventricular fibroblasts as hV-CF.

Growth curves are presented comparing primary hCF (now called NHCF-V) with one line of hPSC-CF, and they conclude that hPSC-CF also become senescent and stop growing after 10 passages: the two curves look quite different and the one for hPSC-CF is still pretty straight, it doesn't seem they have reached the plateau yet so their explanation is not convincing.

Thank you for this observation that caused us to review the data again. We have replotted the cell counts for Fig. 5b after finding an error in calculations for the hPSC-CF. Furthermore, we now have additional passages for the hPSC-CF which show the growth plateauing, please refer to the new Fig. 5b. Although both cell populations show evidence for senescence after multiple passages, the hPSC-CFs are able to be passaged slightly longer consistent with more embryonic-like features.

Fig 3e is slightly random/out of place: it's no longer a comparison with adult fibroblasts but a comparison of the two protocols, GiWi and GiFGF (d20 and after 5 passages), to show again increased expression of the fibroblast marker, almost absent expression of MF20 in the GiFGF protocol and very low level of endothelial cells from both protocols (CD31 for

some reason is plotted differently: histogram instead of pseudo-color plot). This could be integrated into the protocol section.

Thank you for the excellent suggestions. We have revised the figures to plot the old Figure 3e (flow plots comparing GiWi and GiFGF) to the new Figure 4b along with other data comparing GiWi and GiFGF protocols. In accordance with the reviewer's suggestion, we have also changed the data presentation for CD31 to a pseudo-color plot.

4. Fig 4 compares different type of fibroblasts via RNAseq and the 3D ECM scaffold produced by the cells. The message again is that hPSC-CF and NHCF-V cluster together compared to hDF, but no comment is made of the differences among them. For example, the cultures of hPSC-CF are thicker, with more cellular layers than the adult ones (4 versus 2 from the images), this could be another indication of "immature" phenotype, maybe the cells are less susceptible to contact inhibition or just more proliferative, but these differences should be acknowledged.

As the reviewer notes, the hPSC-CFs do generate thicker constructs. We agree that the hPSC-CFs likely exhibit a more embryonic phenotype compared to the adult hV-CFs, which could contribute to the thicker constructs because of potentially reduced contact inhibition. In our revised Discussion we discuss the more embryonic phenotype of the hPSC-CFs and highlight the reviewers point by stating "Additionally, the more robust ECM production and greater cell growth in the high density hPSC-CF cultures compared to adult hV-CFs is consistent with a less mature CF population that may exhibit less contact inhibition of proliferation."

Also, from the heat map it looks like hPSC-CF express more *Nkx2-5*, *WT1*, *Isl1*, *Hey1*, *Hand2* and less *HAND6* (assume *GATA6*) compared to the adult ones, suggesting a more "immature" phenotype?

We appreciate reviewer's close attention to our gene expression data, and we agree that the hPSC-CFs likely have a more 'immature' phenotype. We now directly compare the gene expression patterns of the hPSC-CFs and hV-CFs using gene ontology (GO) analysis for the two cell populations. As stated in the revised results, "Interestingly, genes involved in pattern specification process, anterior/posterior pattern formation, and embryonic morphogenesis are enriched in hPSC-CFs consistent with the hPSC-CFs being more embryonic or immature in phenotype." Furthermore, we have taken advantage of a prior study comparing native mouse embryonic and adult CF gene expression to assess for maturity of the hPSC-CFs by Ieda et al, 2009. Our revised Results now state, "We further examined the subset of differentially expressed genes identified in a prior study comparing embryonic and adult mouse CFs². Mouse embryonic CFs express a subset of ECM genes more highly than adult CFs including *Tnc*, *Fnl1*, *Postn*, *Hapln1*, *Col5a2*, *Col3a1*, and *Col12a2*; and our RNA-seq data also show higher expression in hPSC-CFs than adult hV-CFs of *TNC*, *FNI*, *HAPLN1*, *COL5A2*, *COL3A1*, and *COL12A1* but not *POSTN* (Supplementary Fig. S7a). Mouse embryonic CFs also show differences in cytokines and growth factor expression compared to adult CFs. One of the most important differences is in *Il6* expression which is highly expressed in adult CFs and linked to hypertrophy signaling relative to low levels in mouse embryonic CFs², and our RNA-seq data demonstrated a 52-fold higher expression of *IL6* in the primary adult hV-CF vs hPSC-CF. However, we did not see the higher expression of *HBEGF* and *PTN* in hPSC-CFs compared to adult hV-CFs as was observed in mouse CFs (Supplementary Fig. S7b)."

It is surprising that their differentiated CFs do express WT1 suggesting an epicardial phenotype, and that there is no signal for Tbx20 which was an abundant CF transcript in a previous report (Furtado et al, Circ Res, 2014).

Yes, some level of expression of WT1 is suggested by our RNA-seq data in Fig. 6, but the level is low. Furthermore, when we specifically looked at WT1 expression with qPCR in Fig. 3a, we see minimal expression throughout the time course of the GiFGF protocol in clear contrast to the strong up regulation observed by Iyer et al., and Witty et al. that generate epicardial-derived hPSC-CFs. The combination of trivial expression of WT1 and no expression of TBX18 argue against epicardial-derived fibroblasts. Regarding the lack of Tbx20 expression which has been identified previously in CFs, we directly address this by stating in the revised Discussion, “TBX20 is also expressed in adult mouse and human CFs⁵ which we confirmed in our RNA-seq data for adult hV-CFs; however, minimal TBX20 expression was present in the hPSC-CFs. The lack of TBX20 could reflect the different developmental lineage of these hPSC-CFs or potentially a more immature phenotype.”

Coll and fibronectin expression do not seem to overlap with the cell layer for hPSC-CFs, but do for NHCF-V and hDFs, which is unexplained.

We can appreciate the reviewer’s interpretation based on the merged images, but we would suggest that collagen is expressed across the full thickness based on side view, although to a lesser extent in the lower layer which we state in the revised manuscript.

Note that Gata4 is NOT specific to cardiac fibroblasts; it is expressed by other organ fibroblasts and this should be corrected.

Thank you for pointing out this misstatement. We have corrected the text to state, “The transcription factor GATA4 has previously been reported to be expressed in human CFs but not in DFs⁵...”

5. The data in Fig 5 are very interesting: they measure differences in electrical impulse propagation in mixed cultures of hPSC-CMs with either hPSC-CFs or hDF. For some reason, adult cardiac fibroblasts disappear from this assay, which would have been a good control. The supplementary movies should be relabeled as S1a, S1b, S2a, S2b, it’s not easy to understand what is what.

Thank you for this suggestion. We have completed an additional series of experiments using hV-CF co-culture to add to the prior results. These new data are now presented in the revised Fig. 8 and Supplementary Movies S2b and S3b, and described in the Results. Likewise, we have relabeled the supplementary videos as suggested.

Minor comment:

Apart from a clear inconsistency with the use of acronyms, typos and unclear figure legends, the guidelines for the journal were not followed; for example there shouldn’t be references in the abstract etc.

Thank you for pointing out these issues. The revised manuscript clarifies the acronyms and has revised figure legends for clarity. We also have revised the format to conform to the journal guidelines.

Reviewer #2:

Zhang et al developed the protocol to induce CFs from human PSCs by directed differentiation. Sequential modulation of Wnt and FGF signaling induced second heart field progenitors that efficiently differentiated into hPSC-CFs. However, the cells did not express epicardial genes before differentiation into CFs. They also showed that the hPSC-CFs were similar to native heart CFs than dermal fibroblasts in gene expression and matrix production. Co-culture of hPSC-CFs with CMs demonstrated that hPSC-CFs were different from hDFs in modulation of hPSC-CM electrophysiology. Although the concept is interesting, it remains unclear why the hPSC-CFs did not express epicardial genes before CF differentiation. Moreover, if the authors can show difference between diseased-CFs and healthy-CFs from PSCs, the hPSC-CFs may be useful for disease modeling and drug discovery.

Major Points:

1. Fig 2.

Why the epicardial genes were not upregulated during differentiation? Given that a large population of CFs are derived from the epicardium, it is important to determine the path of CFs in this protocol.

Yes, we completely agree with the reviewer, and we have performed significant additional experiments to address this issue. First, our data argue against an epicardial source for the hPSC-CFs given the lack of expression of classic epicardial markers during differentiation including WT1 and Tbx18 (Fig 3a). This is consistent with the findings of Loh et al. (Cell 2016) that demonstrated that only with inhibition of bFGF signaling in cardiac mesodermal progenitors could (pro)epicardial gene expression be observed. Instead, we suggest that our protocol generates hPSC-CF through the other major pathway in cardiac development, via SHF and endocardial progenitors. We provide strong data showing SHF progenitor intermediates, and we performed additional experiments looking for cells expressing endocardial/endothelial-related genes Tie2 (Fig. 4) and NFATc1 (Suppl Fig. S3). We could not identify a significant population of the endocardial/endothelial intermediates. Nevertheless, our time course could have missed a very transient expression of these markers. Thus, we describe our hPSC-CFs as derived from SHF without emphasizing endocardial intermediates, as we did not clearly identify these.

2. Fig 4, 5

The hPSC-CFs and hCFs resulted in polarized and thick matrix production in contrast to hDFs. What is the mechanism of this difference?

We share the reviewer's interest in the mechanisms of this difference. However, the goal of the experiment was to benchmark the hPSC-CFs relative to the different tissue-specific fibroblast populations based on differential functional responses such as growth properties and ECM production. We were able to demonstrate in this assay that hPSC-CFs performed more similarly to hV-CFs than hDFs, but the mechanistic basis for these differential responses is beyond the scope of this work aimed at characterizing a novel population of hPSC-CFs.

Moreover, it would be important to determine the molecular mechanism for different electrophysiology in co-cultured hPSC-CMs between hPSC-CFs and hDFs.

We are extraordinarily excited to examine the mechanistic basis of the differential effects of co-culture on the cardiac electrophysiology. Given the decades long debates about the role of CFs in cardiac electrophysiology, there are opportunities to provide important new mechanistic insights.

However, definitive studies parsing out the mechanisms are complex and require a range of experiments and techniques that are beyond the scope of this initial paper identifying a novel hPSC-CF population.

3. Fig 5

Fibroblasts change the property by sequential passages and culture conditions. It remains unclear whether the hPSC-CFs are truly useful for disease modeling and drug discovery. Demonstrating the difference between diseased-CFs and healthy-CFs from human PSCs may be informative.

We agree and look forward to future experiments to investigate diseases modeling with the hPSC-CF. The reviewer clearly identifies one of the opportunities these cells present for future studies. We acknowledge that cell culture conditions and passaging are important variables in the study of CFs, but this should not eliminate the possibility of informative studies.

Reviewer #3 (Remarks to the Author):

The central objective of this manuscript was to develop an efficient method for obtaining cardiac fibroblasts from human PSCs by manipulating Wnt and FGF signaling. The authors strategy was to form cardiac mesodermal progenitors through routine inhibition of Wnt signaling as done for inducing myocyte differentiation and tested whether FGF stimulation could bias the differentiation of these progenitors to non-myocyte cell types. After examination of dose-response curves with recombinant FGF treatment the authors had made spindle shaped cells that secrete extracellular matrix and express markers associated with cardiac fibroblasts like Vimentin, Perisotin, FSP1 and a little TCF21. Co-cultures with the various fibroblasts and myocytes demonstrated that the type and quantity of fibroblasts in co-culture remodel the myocytes action potential. The impact of the paper is potentially high as the ability to have a human model of fibroblast-based disease, as there is a growing need to find ways of treating cardiac fibrosis. The technology is also novel and could help pioneer new developments in the field, so this paper is of tremendous value. The manuscript does fall short in proving that the GiFGF protocol is really making cardiac fibroblasts, and what features of cardiac fibroblasts the cells are truly modeling.

- The biggest problem with the paper, which is a problem for the field in general, is the genetic heterogeneity of fibroblasts within the heart and what genetic signatures correlate with given fibroblast functions. The authors make a decent attempt at trying to comparing the GiFGF fibroblasts to the Lonza human cardiac fibroblast cell line, but this particular line is a mixed population of valve, ventricular, and atrial fibroblasts which we know at least in the mouse have very different genetic signatures and functions, making it challenging to know which fibroblast population the GiFGF fibroblasts actually models.

We agree with the reviewer that the heterogeneity of ‘fibroblast’ populations is a challenge for the field with distinct genetic signatures. Nevertheless, this should only motivate further efforts to address this limitation, and perhaps hPSCs-related experiments will provide new insights into

the heterogeneity in the human system. We did try to minimize the variability in choosing the primary hV-CF which were from Lonza (NHCF-V, normal heart cardiac fibroblast-ventricle) and are derived from ventricle without valve or atrial components. This is based on the description of the fibroblast source and confirmed by personal communication with the Lonza technical representatives. So we cannot exclude important genetic heterogeneity, but at least our comparator native heart hCF comes from ventricle.

- Moreover, the genetic signatures of these Day 20 fibroblasts have some overlap with well-characterized mouse ventricle populations (ie. expression of vimentin, periostin, a little TCF21) yet they don't completely match cardiac fibroblast signatures either.

We agree that our hPSC-CF population is not a perfect match for adult hV-CFs. However, they are a closely related cell population that would be anticipated to show differences based on the heterogeneity of the CFs found in the adult human ventricle relative to a more restricted developmental pathways for the hPSC-CFs as well as the relative immaturity of the hPSC-CF similar to other in vitro differentiated lineages from hPSCs.

In addition, the co-culture experiments while interesting and important are too early and don't fit the papers context. It would be better to spend that experimental space proving the GiFGF fibroblasts truly function like cardiac fibroblasts by testing whether they respond similarly to drugs like TGF β or Angiotensin II, or determining what state of maturation these fibroblasts are in.

We respectfully disagree with the reviewer regarding the co-culture experiments as out of place because they provide an important comparative assay of CF function. Although we do agree that detailed mechanistic studies of the differential responses remains a better topic for future research. Thank you for the suggestion of testing TGF β effects on the population, which are now shown in revised Figure 7C. We indeed find that the hV-CFs and hPSC-CFs respond similarly to 48 hours of TGF β 1 stimulation exhibiting features of myofibroblast transformation including actin stress fiber formation.

We have made additional effort to characterize the maturation state of the hPSC-CFs, based on the pattern of gene expression with a new gene ontology analysis in Fig. 6d comparing the populations. In the revised manuscript results we now state, “Interestingly, genes involved in pattern specification process, anterior/posterior pattern formation, and embryonic morphogenesis are enriched in hPSC-CFs consistent with the hPSC-CFs being more embryonic or immature in phenotype.” Furthermore, we have taken advantage of a prior study comparing native mouse embryonic and adult CF gene expression to assess for maturity of the hPSC-CFs by Ieda et al, 2009. Our revised results now state, “We further examined the subset of differentially expressed genes identified in a prior study comparing embryonic and adult mouse CFs². Mouse embryonic CFs express a subset of ECM genes more highly than adult CFs including *Tnc*, *Fn1*, *Postn*, *Hapln1*, *Col5a2*, *Col3a1*, and *Col12a2*, and our RNA-seq data also showed the relatively higher expression in hPSC-CFs than adult hV-CFs of *TNC*, *FNI*, *HAPLN1*, *COL5A2*, *COL3A1*, and *COL12A1* but not for *POSTN* (Supplementary Fig. S7a). Mouse embryonic CFs also show differences in cytokines and growth factor expression compared to adult CFs. One of the most

important differences is in *Il6* expression which is highly expressed in adult CFs and linked to hypertrophy signaling relative to low levels in mouse embryonic CFs², and our RNA-seq data demonstrated a 51-fold higher expression of *IL6* in the primary adult hV-CF vs hPSC-CF. However, we did not see the higher expression of HBEGF and PTN in hPSC-CFs compared to adult hV-CFs as was observed in mouse CFs (Supplementary Fig. S7b).”

- Finally, the authors claim efficient induction of the cardiac fibroblast phenotype which is based on flow sorting with a fibroblast surface marker TE-7; however, the specificity of TE-7 is questionable as many fibroblast surface markers also bind to dendritic cells (CD11b) or other hematopoietic cells. Validation of the purity of the cell population labeled by TE-7 would be helpful towards understanding the efficiency of GiFGF-mediated differentiation.

Thank you for this suggestion because we agree that the more markers we can use to define the population, the better. Our experience with the TE-7 antibody over the years has been favorable for a fibroblast specific antibody, but to support this conclusion, we provide new data in Figure 5d and Figure S6 showing that more than 97% of TE-7 labeled hPSC-CFs are positive for intermediate filament protein, vimentin, which is typically present in fibroblasts. Additionally in Figure 5e, we found 90% of the TE-7 positive cells are positive for PDGFR α like native heart sources. This complements the immunolabeling experiments also showing FSP1 labeling the vast majority of cells in Fig. 5h.

Below are additional comments that also complicate the interpretation of this cardiac fibroblast differentiation protocol.

- In combining the expression data for the GiFGF induced fibroblast differentiation these cells seemed to be labeled by a little TCF21, Vimentin, Periostin, and FSP1 gene expression. Recent lineage tracing experiments have demonstrated that in adult cardiac fibroblasts Periostin only turns on with injury and that FSP1 actually labels hematopoietic cells. Moreover, Periostin gene expression has been reported in non-fibroblast cells when they are grown in culture. This is of concern given that traditional epicardial lineage marks aren't expressed nor were data provided to suggest they are more like endothelial derived cardiac fibroblasts that have Tie2 or NfactC1 expression. As stated these cells look like they recapitulate the second heart field development but whether fibroblasts in that region have different genetic signatures and functions is not well understood. So again its unclear what this GiFGF population is modeling. Can the authors identify additional markers like PDGFR α or Hey1 that might add support for these cells being cardiac fibroblasts? Or, the authors could compare these cells to fibroblasts obtained from the second heart field.

Thank you for the thoughtful insights into the possible developmental pathways contributing to the generation of hPSC-CFs. Regarding POSTN, we do find some expression based on Q-PCR in our hPSC-CFs which is to be expected in the developing heart based on prior lineage tracing studies (see Snider et al., 2009, Circ Res 105:934). To better address the endocardial intermediates we have now done flow cytometry for Tie2 and NFATc1 and not observed significant cell populations positive for these markers. It is possible that they are transiently

present and missed by our sampling, so we have focused on SHF progenitors as we have the strongest evidence for these intermediates. As the reviewer suggests, we have looked at PDGFR α and found that 90% of the hPSC-CF are positive for this surface maker (Fig 5e). HEY1 has been examined at the gene expression level in the RNA-seq data and was highly expressed in hPSC-CFs (Fig 6b). Unfortunately, we do not have a good comparator population of SHF-derived CFs from native human embryonic heart given limited accessibility and challenges in identifying of SHF-specific CF population in the native human heart.

- It's clear these cells are making matrix but that can happen with many cell populations once they get into culture. Can the authors provide additional functional data that could prove these cells behave like cardiac fibroblasts? As stated, above perhaps looking at function with respect to their response to mechanical or chemical stimuli could answer this question.

Thank you for this suggestion. We have now examined the well-studied effect of TGF β on CFs shown in revised Fig. 7c. We indeed find that the hV-CFs and hPSC-CFs respond similarly to 48 hours of TGF β 1 stimulation exhibiting features of myofibroblast transformation including actin stress fiber formation.

- The authors need to add other fibroblasts populations as a basis for comparison given that the Lonza cell and dermal cells while human don't give enough specificity to the ventricle. Ideally the authors could get some human ventricular fibroblasts but even comparisons to neonatal and adult mouse ventricular cardiac fibroblasts would help with anatomical specificity and to define the fibroblasts stage of maturation. This may also help sort out why these fibroblasts are expressing periostin as this is more of the profile for neonates or injured hearts.

We apologize that the origin of the Lonza CFs was not clear. The Lonza cells are normal heart cardiac fibroblasts from ventricle (NHCF-V) based on the Lonza product description and our confirming discussion with technical representatives. We have also added another control of human adult CFs isolated from LV (F1-V) by our co-author (EGS) at UW in the characterization of the hPSC-CFs (Fig. 5c) which perform similarly.

- The authors should validate the TE-7 antibody and show that it only labels cardiac fibroblasts.

Thank you for this suggestion. Validating the TE-7 antibody in native human heart tissue is logical, but the challenge remains how to clearly identify the CF populations in the human heart with other existing markers and lack of genetic tracing which has proven so valuable in mouse studies. Based on the additional markers used to support the ability of TE-7 to identify fibroblasts in vitro including co-labeling with PDGFR α , vimentin, FSP1, we suggest there is good evidence for specificity of this antibody.

Minor comments

- An important consideration in examining markers and fibroblast function are whether these fibroblasts convert to myofibroblasts with classic triggers and why these fibroblasts aren't activating during the 20 days of culturing on plastic.

Yes, this is a major challenge in culturing and studying CFs. We have directly examined for myofibroblasts in our cell preparations in the revised Fig. 7c. In the 20 day differentiated hPSC-CFs using the defined medium of CFBM plus 75ng/ml bFGF, as well as during subsequent passaging with FibroGro medium plus 2% FBS for hPSC-CFs or FGM-3 medium plus 10% FBS for hV-CFs, we see only low levels of SMA expression as shown in Fig. 7c (confluent p5 as the control). However, when the hPSC-CFs and hV-CFs are cultured in a basal medium of DMEM with 10% FBS, a high SMA expressing population of CFs rapidly emerges. The size of the SMA high population further increases with addition of TGF β 1 and SMA-positive stress fibers also become evident consistent with a myofibroblast phenotype.

Reviewers' comments:

Reviewer #1 (Remarks to the Author):

Zhang et al. have optimized a protocol to obtain mesenchymal, collagen-producing cells from human pluripotent stem cells through generation of second heart field progenitors, mediated by Wnt and FGF signaling.

In the revised version of the manuscript, the authors explain more clearly the relevance of their work in the introduction, as an alternative way to obtain cardiac fibroblasts, since the primary patient-derived cells undergo senescence with passaging. However, in figure 5b and at page 10 they show that "The hPSC-CFs can propagate for more than 10 passages before undergoing senescence similar to primary hV-CFs", putting in doubt the relevance of their claim. This needs clarification.

The quality of the manuscript is significantly improved with more attention to the use of acronyms, better presentation of data and re-organization of figures. However, some concern remains on the identity of the presented cells. The array experiment shows that hPSC-CF are more similar to hV-CF than hDF, but still more immature. To confirm the more immature phenotype, the authors refer to the paper published by Ieda et al. and compare the ratio of gene expression between hPSC-CFs/ hV-CFs with the previously published ratio of embryonic/adult CFs, concluding that "gene expression data are consistent with hPSC-CFs being most closely related to hV-CFs relative to the other cell types tested, but the gene expression pattern is consistent with the hPSC-CFs having a more immature phenotype". The absence of Tbx20 expression and very low level of Nkx2-5 does not support claims of an immature phenotype. This needs clarification.

There are a few additional differences with the data in Ieda et al that are not well discussed in the manuscript. For example, hPSC-CFs express higher levels of CSF1, BMP2 and the extracellular proteins TNC, LUM, NID1, compared to both hV-CFs and embryonic-CFs. The possibility cannot be excluded that the observed differences are due to the enrichment of a particular subset of fibroblasts in areas where SHF contribution is more significant (OFT, right atria and ventricle), distinct from the main epicardial-derived fibroblast population. This needs discussion.

Since the comparison with cardiac fibroblasts from hPSC-derived epicardial cells in vitro is considered beyond the scope of this paper, commercially available human embryonic fibroblasts could be used as additional control for the profiling and functional assays (Fig. 5, 7, 8).

Minor comments:

Regarding the TGFb assay in figure 7: starting the treatment with cells at lower density could help with discerning the typical myofibroblast morphology. Please include hDFs as control.

Reviewer #2 (Remarks to the Author):

The authors tested the route of hPSC-CF derivation from SHF by endocardial and endothelial marker expression. The cells did not express these genes nor epicardial markers. Thus, the routes of these CFs are obscure.

As the authors could not respond to my comments (#1, 2) properly, I am not sure whether their hPSC-CFs are bona-fide CFs and this paper could contribute significantly to this field.

Reviewer #3 (Remarks to the Author):

This is the second review of the manuscript by Zhang et al. Overall the authors did an adequate job responding to the reviewer's criticism, and the manuscript is substantially improved. A major criticism of the paper that still remains is that the authors provide no direct comparison to a fetal or neonatal fibroblast in terms of in vitro culture thickness, morphology, activation, or cardiomyocyte pacing, although the data that is presented provide sound premise for their proposed hypothesis that hPSC-CFs resemble an immature fibroblast phenotype. It is acknowledged that getting the appropriate human fibroblast samples is a challenge and perhaps comparing it directly to murine developing fibroblasts may be the closest benchmarks for these comparisons. Moreover, the faster spontaneous pacing of CMs by hPSC-CFs compared to hvCFs could provide additional evidence for an immature phenotype, and would benefit from a direct comparison to commercially available human fetal cardiac fibroblasts. Again a fetal/neonatal rodent fibroblasts comparison could be useful here, especially considering the comparison to embryonic mouse CF is already made for ECM gene expression data. Despite these remaining criticisms, the manuscript addresses an important need in the field that will help move our understanding of cardiac fibroblast biology forward, and thus is ready for publication. There are also some minor concerns that should be addressed prior to publication including:

- The authors state that CFs, " (comprise) a significant fraction of cells in the heart estimated using current lineage tracing techniques to be about 20% of the nonmyocyte cells in the mouse heart," citing a paper by Pinto et al. which finds CFs to comprise only 15% of nonmyocytes in the heart (See figure 5 in Pinto et al.).
- The authors use TE7 antibody to gate for fibroblasts in their endpoint flow cytometry analyses but do not present any data on the percentage of cells that are TE7+ through the course of the differentiation protocol.
- On page 8 the description of vimentin's relevance with respect to fibroblasts is correct, but on page 10 the authors refer to vimentin as a "general fibroblast marker," when it is not specific to fibroblasts and instead indicates the epithelial-mesenchymal transition.
- On page 11 the authors claim that no cTnT positive cardiomyocytes were detected in any fibroblast lines, but do not present the requisite controls (secondary antibody only, cardiomyocyte positive) to back that claim up effectively.
- To bolster the cardiac specificity of the experiment involving fibroblast activation (Figure 5c), angiotensin II should be used in addition to TGF-beta as an activating agent. Periostin staining/expression in addition to aSMA is an important secondary marker of the ability of these cells to activate in response to known agonists.
- In addition for Figure 5a, the authors make the claim that hDF matrix production is uniform but hv-CF and hPSC-CF matrix is "polarized" with fibronectin at the top. Since the CFs are multilayered, the DFs are not, and the staining protocol used does not permeabilize the cells, this finding could simply be an artifact of poor antibody penetration into the lower cell layers. Discussion of literature to give biological relevance to the observation of "polarized matrix" would be very helpful here.

RESPONSE TO REVIEWER #1

“In the revised version of the manuscript, the authors explain more clearly the relevance of their work in the introduction, as an alternative way to obtain cardiac fibroblasts, since the primary patient-derived cells undergo senescence with passaging. However, in Figure 5b and at page 10 they show that “The hPSC-CFs can propagate for more than 10 passages before undergoing senescence similar to primary hV-CFs”, putting in doubt the relevance of their claim. This needs clarification.”

Thank you for the comment demonstrating that we have improved the description of our rationale, but it still lacked clarity. We agree with you that both primary hV-CFs and hPSC-CFs undergo senescence with sequential passaging; however, there is an unlimited source of hPSCs to derive more genetically identical hPSC-CFs in contrast to the limited supply of genetically identical cardiac tissue-derived CFs. Moreover, many disease- and patient-specific heart tissue samples are not accessible, which impedes studies of human CF biology and their role in disease.

The revised manuscript states. “However, disease- and patient-specific cardiac tissue samples are not always available, for example, many patients with inherited heart disease typically do not undergo cardiac surgical interventions. Thus, a robust, reproducible, and patient-specific source of CFs is desirable. Because hPSCs can renew indefinitely in culture, hPSCs provide an unlimited source of genetically identical hPSC-CFs in contrast to primary cardiac tissue-derived CFs.”

The quality of the manuscript is significantly improved with more attention to the use of acronyms, better presentation of data and re-organization of figures. However, some concern remains on the identity of the presented cells. The array experiment shows that hPSC-CF are more similar to hV-CF than hDF, but still more immature. To confirm the more immature phenotype, the authors refer to the paper published by Ieda et al. and compare the ratio of gene expression between hPSC-CFs/ hV-CFs with the previously published ratio of embryonic/adult CFs, concluding that “gene expression data are consistent with hPSC-CFs being most closely related to hV-CFs relative to the other cell types tested, but the gene expression pattern is consistent with the hPSC-CFs having a more immature phenotype”. The absence of Tbx20 expression and very low level of Nkx2-5 does not support claims of an immature phenotype. This needs clarification.

Thank you for this important question regarding the phenotypic maturity of the hPSC-CFs. The question motivated us to review the literature in more depth and perform additional experiments with purchased primary human fetal CFs for comparison. We now base our gene expression comparisons with a recent publication by Jonsson et al.¹ which specifically compares RNA-seq-based gene expression between primary 21-week gestation fetal and adult human ventricular CFs rather than Ieda et al., which used a genetically engineered mouse model. Furthermore, the Ieda et al., gene expression data is difficult to directly compare with our data because the fetal and adult CF data from Ieda et al. are normalized to embryonic Nkx2.5-YFP positive cells (defined as cardiomyocytes). Because Nkx2.5 is expressed in embryonic CFs based on the Jonsson study (Appendix 2), such a normalization strategy could bias the results to not represent the complete population of CFs in the embryonic mouse heart. Given the more comparable human RNA-seq datasets from Jonsson et al. and our study, we have now used those data as a primary comparator and removed the previous detailed comparison to the mouse data in Ieda et al. This more direct comparison with our hPSC-CFs demonstrates the hPSC-CFs are more similar to human fetal ventricular CFs (hfV-CFs) than to human adult ventricular CFs (haV-CFs) (see Suppl. Fig. S8) consistent with a more embryonic phenotype of hPSC-CFs. Regarding TBX20 expression, it is expressed in haV-CFs, but not detectable in the hPSC-CFs (see our Suppl Appendix 2). A similar pattern of expression was observed in the Jonsson study (see Jonsson’s Appendix 2). Regarding NKX2-5 expression, it is expressed in hPSC-CFs but not detectable in haV-CFs (see our Suppl Appendix 1), again in agreement with the data of Jonsson et al., (see Jonsson’s Appendix 2). In addition, we have performed significant additional experiments adding human fetal ventricular CFs for comparison with hPSC-CFs, human adult CFs, and hDFs. The new data are presented in Figures 5, 6, 8, and Supplementary Figures S4, S5, S7, S8, S10. In the range of assays tested, the hPSC-CFs are most similar to hfV-CFs suggesting a more embryonic phenotype. Nevertheless, the issue of immaturity with regard to CFs is complex with multiple distinct origins of fibroblasts in the heart which may change in their relative abundances during heart development. Thus assuming a simple linear maturation process is an oversimplification. We identify this limitation in our data interpretation in the Discussion in revised manuscript, “Although these functional

assays and gene expression studies are consistent with hPSC-CFs having a more embryonic than adult phenotype, there are other possible explanations for these differences given the multiple sources of CFs that may exhibit differences in relative abundance in fetal and adult human heart.”

There are a few additional differences with the data in Ieda et al that are not well discussed in the manuscript. For example, hPSC-CFs express higher levels of CSF1, BMP2 and the extracellular proteins TNC, LUM, NID1, compared to both hV-CFs and embryonic-CFs. The possibility cannot be excluded that the observed differences are due to the enrichment of a particular subset of fibroblasts in areas where SHF contribution is more significant (OFT, right atria and ventricle), distinct from the main epicardial-derived fibroblast population. This needs discussion.

We appreciate the reviewer’s concern regarding the difference of the gene expression when compared our hPSC-CF vs. haV-CF with Ieda’s mouse embryonic CF vs. adult CF gene expression data. As we discussed above, further consideration of the Ieda’s data and the availability of a more relevant comparative human dataset has prompted us to revise this section of the manuscript. We have performed new bioinformatics analysis of our RNA-seq data of hPSC-CF, haV-CF, and hDF combined with the RNA-seq data of hfV-CF and haV-CFs from Jonsson et al. with the results are presented in the new Supplementary Fig. S8. This analysis suggests the hPSC-CFs are most similar to the hfV-CFs. Nevertheless, we agree with the reviewer that the differences observed could be in part due to enrichment of a particular subset of SHF-related CFs. At this time, we do not have adequately defined native heart samples to address this possibility. For example, the primary CF controls we have used in our study, including hfV-CFs from Cell Applications are derived from ventricle without further definition of right or left, and the haV-CFs from Lonza are derived from left ventricle. Thus, we could only conclude that the hPSC-CFs are more similar to hfV-CFs than haV-CFs. However, the reviewer raises an important question for future studies.

Since the comparison with cardiac fibroblasts from hPSC-derived epicardial cells in vitro is considered beyond the scope of this paper, commercially available human embryonic fibroblasts could be used as additional control for the profiling and functional assays (Fig. 5, 7, 8).

Thank you for this excellent suggestion. We purchased commercially available primary human fetal ventricular cardiac fibroblasts (hfV-CF) from Cell Applications, Inc. (San Diego, CA, USA) and have performed additional experiments to compare these cells with the hPSC-CF as well as with haV-CF and hDF. New data are presented in the new Fig. 5, 6, 8, and Supplementary Fig. S4, S5, S7, S8, and S10. These new data show that the phenotypic properties of hPSC-CFs are most similar to hfV-CFs such as cell proliferation (Fig. 5), fibroblast markers expression (Fig. 6) and ECM production and matrix thickness (Fig 8). Regarding cardiomyocyte electrophysiology in the co-culture assay, we were not able to perform additional experiments with the hfV-CFs in a timely fashion, and these experiments are an important area for future research. Overall, the profiling studies and the gene expression pattern (using human fetal CF data from Jonsson et al., for comparison) provide more substantial support for our conclusion that the hPSC-CF exhibit a more embryonic phenotype.

Minor comments:

Regarding the TGFb assay in figure 7: starting the treatment with cells at lower density could help with discerning the typical myofibroblast morphology. Please include hDFs as control.

Thank you for these suggestions. We agree that based on morphology alone, lower density cultures can help reveal myofibroblast morphology, but we preferred to maintain our cell culture conditions given we have evidence for myofibroblast transformation that is quantitative using flow cytometry as well as with detection of SMA stress fibers with immunolabeling. As suggested, we have also included the hDFs in the myofibroblast induction experiments. The hDFs perform differently in this assay compared to all of the CF populations tested. The new data are presented in the new Fig. 8c and Supplementary Fig. S10.

RESPONSE TO REVIEWER 2

The authors tested the route of hPSC-CF derivation from SHF by endocardial and endothelial marker expression. The cells did not express these genes nor epicardial markers. Thus, the routes of these CFs are obscure.

Thank you for this comment. Yes, we did not find evidence for clear endocardial or epicardial intermediates; however, our working model is that the SHF progenitors can directly give rise to CFs. This is comparable to the findings of Milgrom-Hoffman et al. that demonstrate that a subset of endocardial cells are derived from the SHF independently of endocardial/endothelial progenitor cells². Such multipotent SHF progenitors have been reported recently by others in vivo and in vitro which give rise to fibroblast populations as well³.

As the authors could not respond to my comments (#1, 2) properly, I am not sure whether their hPSC-CFs are bona-fide CFs and this paper could contribute significantly to this field.

Regarding comment #2 from the prior review (The hPSC-CFs and hCFs resulted in polarized and thick matrix production in contrast to hDFs. What is the mechanism of this difference?), we provide this comparison to focus on comparative properties of different fibroblast populations. We suggest that the thick matrix of CF cultures is due to the fact that the hDF under our conditions show contact inhibition and do not grow as multi-layer cultures like the CFs including the hPSC-CFs, the primary human fetal and adult CFs which produce thicker matrix. Regarding polarized matrix with fibronectin most abundant in the upper layer, we have performed further experiments to examine fibronectin in both non-permeabilized and permeabilized hPSC-CF cultures. We present these data in the revised manuscript on page 14 and 15 as “To examine the ECM production and assembly, the fibroblast cultures were fixed, but not permeabilized, and immunolabeled with the antibodies to collagen type I and human fibronectin. The nuclei were stained with Hoechst. Confocal imaging demonstrated that all of the fibroblast preparations robustly produced collagen I and fibronectin (**Fig. 8a**). However, 3D reconstruction of the confocal images revealed that the hPSC-CFs, hfV-CFs and haV-CFs produced multicell layer cultures while the hDFs produced only monolayer cultures (**Fig. 8a, Supplementary Movie S1, S2, S3, S4**). The average depths of the multilayer hPSC-CF (~40 μm) and the hfV-CF (~44 μm) cultures were similar and greater than the haV-CF cultures (~24 μm) and the monolayer hDF

cultures (~12.5 μm) (**Fig. 8b**). Although extracellular fibronectin and collagen I were evenly distributed throughout the monolayer hDF cultures, fibronectin was concentrated in the top layers of all of the CF cultures in contrast to the distribution of collagen I throughout the depth of the multilayer CF cultures (**Fig. 8a**). To confirm adequate antibody penetration and ECM protein labeling, multilayer hPSC-CF cultures were permeabilized before immunolabeling. The permeabilized hPSC-CF culture shows a distinct pattern of fibronectin immunolabeling with clear intracellular fibronectin detected as well as more fibrillar extracellular fibronectin concentrated in the upper layers (**Supplementary Fig. S9, Movie S5**). Polarization of extracellular fibronectin has been observed in other *in vitro* cultured cell systems and attributed to the binding cell produced fibronectin with soluble fibronectin in the serum containing medium to generate insoluble fibronectin concentrated at the apical surface of the cultures^{4,5,6}.”

RESPONSE TO REVIEWER #3

This is the second review of the manuscript by Zhang et al. Overall the authors did an adequate job responding to the reviewer’s criticism, and the manuscript is substantially improved. A major criticism of the paper that still remains is that the authors provide no direct comparison to a fetal or neonatal fibroblast in terms of *in vitro* culture thickness, morphology, activation, or cardiomyocyte pacing, although the data that is presented provide sound premise for their proposed hypothesis that hPSC-CFs resemble an immature fibroblast phenotype. It is acknowledged that getting the appropriate human fibroblast samples is a challenge and perhaps comparing it directly to murine developing fibroblasts may be the closest benchmarks for these comparisons. Moreover, the faster spontaneous pacing of CMs by hPSC-CFs compared to hvCFs could provide additional evidence for an immature phenotype, and would benefit from a direct comparison to commercially available human fetal cardiac fibroblasts. Again a fetal/neonatal rodent fibroblasts comparison could be useful here, especially considering the comparison to embryonic mouse CF is already made for ECM gene expression data. Despite these remaining criticisms, the manuscript addresses an important need in the field that will help move our understanding of cardiac fibroblast biology forward, and thus is ready for publication.

We thank the reviewer for the comments and statement that ‘manuscript addresses an important need in the field...’ We agree. Furthermore, we have been able to improve the manuscript by using a commercially available source of human fetal ventricular cardiac fibroblasts (hfV-CFs) from Cell Applications, Inc. to incorporate in most of characterization assays. The new data are presented in the new Fig. 5, 6, 8, and Supplementary Fig. S4, S5, S7, S8, S10. These new data show that the phenotypic properties of hPSC-CFs are most similar to hfV-CFs such as cell proliferation (Fig. 5), fibroblast markers expression (Fig. 6) and ECM production and matrix thickness (Fig 8). Furthermore, we have been better able to benchmark our gene expression RNA-seq data with the human fetal and adult CF RNA-seq dataset recently published that also supports a more embryonic or immature phenotype of the hPSC-CFs¹ (Supplementary Fig. S8). Regarding cardiomyocyte electrophysiology, we were not able to perform additional experiments with the hfV-CFs in a timely fashion, and these experiments are an important area for future research. Overall, the profiling studies and the gene expression pattern provide more substantial

support for our conclusion that the hPSC-CF exhibit a more embryonic phenotype. Although the issue of immaturity with regard to CFs is likely complex with multiple distinct origins of fibroblasts in the heart which may change in their relative abundances during heart development. Thus assuming a simple linear maturation process is an oversimplification which we acknowledge in the manuscript.

There are also some minor concerns that should be addressed prior to publication including:

- The authors state that CFs, " (comprise) a significant fraction of cells in the heart estimated using current lineage tracing techniques to be about 20% of the nonmyocyte cells in the mouse heart," citing a paper by Pinto et al. which finds CFs to comprise only 15% of nonmyocytes in the heart (See figure 5 in Pinto et al.).

Thank you for the clarification. We have revised the manuscript to state on page 3, line 45, "... to be about 15% of the nonmyocyte cells in mouse heart."

- The authors use TE7 antibody to gate for fibroblasts in their endpoint flow cytometry analyses but do not present any data on the percentage of cells that are TE7+ through the course of the differentiation protocol.

Please see the new Fig. 4d for the intermediate cell populations of TE7 antibody labeling in the time course of the GiFGF protocol.

- On page 8 the description of vimentin's relevance with respect to fibroblasts is correct, but on page 10 the authors refer to vimentin as a "general fibroblast marker," when it is not specific to fibroblasts and instead indicates the epithelial-mesenchymal transition.

We agree and thank you for identifying the misstatement. We have revised the manuscript to state on page 11, line 222, "We tested labeling vimentin, an intermediate filament protein expressed in fibroblasts...."

- On page 11 the authors claim that no cTnT positive cardiomyocytes were detected in any fibroblast lines, but do not present the requisite controls (secondary antibody only, cardiomyocyte positive) to back that claim up effectively.

Please see the new Fig. 6c for secondary antibody only and hPSC-CM positive controls.

- To bolster the cardiac specificity of the experiment involving fibroblast activation (Figure 5c), angiotensin II should be used in addition to TGF-beta as an activating agent. Periostin staining/expression in addition to α SMA is an important secondary marker of the ability of these cells to activate in response to known agonists.

We have performed additional experiments examining the effect of angiotensin II on α SMA expression by hPSC-CFs, hfV-CFs and hDFs using flow cytometry and immunolabeling. The new data are presented in the new Supplementary Fig. S10. Unfortunately, with our assays, we cannot accurately measure periostin expression as it is a secreted extracellular matrix protein that will

not be accurately measured using isolated cells in flow cytometry or readily quantitated by immunofluorescence, which were the two assays used. So we have not included periostin as part of our myofibroblast evaluation. The challenge of measuring changes in periostin expression using antibody-based approaches has also been identified by others in the literature (Kanisicak et al., 2016, Nat. Comm.)

• In addition for Figure 5a, the authors make the claim that hDF matrix production is uniform but hV-CF and hPSC-CF matrix is "polarized" with fibronectin at the top. Since the CFs are multilayered, the DFs are not, and the staining protocol used does not permeabilize the cells, this finding could simply be an artifact of poor antibody penetration into the lower cell layers. Discussion of literature to give biological relevance to the observation of "polarized matrix" would be very helpful here.

Thank you for this question. Yes, the multilayered culture could present a challenge in antibody accessibility, but the purpose of our experiment was to evaluate specifically extracellular matrix proteins focusing on the dominant collagen I and fibronectin proteins. We suggest that the antibodies can access more than superficial layers of the culture because the immunolabeling for collagen was evident throughout the culture and clearly below the fibronectin labeling (Fig. 8a, Supplementary Movie S1, S2, S3). However, to directly address this concern, we performed immunolabeling on permeabilized hPSC-CF high density cultures (Supplementary Fig. S9, Movie S5). Fibronectin immunolabeling was seen more diffusely, but it was concentrated in intracellular regions for lower layers of the cultures and only in fibrillar structures in the upper layer of the culture. We particularly encourage the reviewer to view the Supplementary Movie S5 of a Z-scan of confocal images to most clearly observe intracellular distribution pattern of fibronectin in lower layers and fibrillar fibronectin in upper layers. Furthermore, in the revised manuscript, we provide movies of the Z-scan confocal images for all of the conditions tested to provide the opportunity for more detailed observation. We also provide references on the biological relevance to the "polarized matrix" in the revised manuscript based on the literature as follows: "Polarization of extracellular fibronectin has been observed in other in vitro cultured cell systems and attributed to the binding cell produced fibronectin with soluble fibronectin in serum containing medium to generate insoluble fibronectin concentrated at the apical surface of the cultures⁴⁻⁶."

Overall, we are delighted to have the opportunity to revise and improve our manuscript benefiting from the reviewers' suggestions and comments. Thank you for the time and effort to review our manuscript.

REFERENCES

1. Jonsson, M.K.B. *et al.* A Transcriptomic and Epigenomic Comparison of Fetal and Adult Human Cardiac Fibroblasts Reveals Novel Key Transcription Factors in Adult Cardiac Fibroblasts. *JACC Basic Transl Sci* **1**, 590-602 (2016).

2. Milgrom-Hoffman, M. *et al.* The heart endocardium is derived from vascular endothelial progenitors. *Development* **138**, 4777-4787 (2011).
3. Andersen, P. *et al.* Precardiac organoids form two heart fields via Bmp/Wnt signaling. *Nat Commun* **9**, 3140 (2018).
4. Kowalczyk, A.P., Tulloh, R.H. & McKeown-Longo, P.J. Polarized fibronectin secretion and localized matrix assembly sites correlate with subendothelial matrix formation. *Blood* **75**, 2335-2342 (1990).
5. McKeown-Longo, P.J. & Mosher, D.F. Binding of plasma fibronectin to cell layers of human skin fibroblasts. *J Cell Biol* **97**, 466-472 (1983).
6. McKeown-Longo, P.J. & Mosher, D.F. Interaction of the 70,000-mol-wt amino-terminal fragment of fibronectin with the matrix-assembly receptor of fibroblasts. *J Cell Biol* **100**, 364-374 (1985).

REVIEWERS' COMMENTS:

Reviewer #1 (Remarks to the Author):

The authors have replied adequately to the reviewers' comments in their second revision. They have explained more clearly what is the rationale and why it is important to have an "off-the-shelf" source of cardiac fibroblasts (line 46) and they added primary human fetal and adult fibroblasts as control in most of the assays, except for the electrophysiology experiment (Fig.9) where additional tests with fetal fibroblasts couldn't be performed in a "timely fashion". As for the gene expression, they removed the comparison with Ieda's study (on murine adult and fetal fibroblasts) and referred to a more recent publication from Jonsson et al. on human fibroblasts.

They claim to have found an efficient protocol to obtain CF from hPSCs through SHF progenitors, but all their data show a very immature phenotype and we don't know how these cells compare to the fibroblasts that can be obtained from hPSCs through epicardial progenitors (the experiment was proposed in the first revision and considered beyond the scope of this manuscript). To justify their alternate approach, in the introduction they mention the three main sources of fibroblasts and cite a relatively old paper (13.Cai, C.L. et al. A myocardial lineage derives from Tbx18 epicardial cells. *Nature* 454,669 104-108 (2008)) to support the idea that just 30% of the fibroblasts are epicardially derived; however the more recent lineage tracing by Moore-Morris T et al, (*JCI* 2014) Fig3 shows that 80% of fibroblasts are epicardial derived and 20% endocardial derived. This should be cited.

Specific comments

Regarding the RNAseq comparison in Fig.7: from the dendrograms and the heat maps it looks like the hPSCCF are relatively more similar to haV CF (adult ventricular) than to hPSC, hPSC-derived cardiomyocytes, and dermal fibroblasts. The comparison with fetal fibroblasts (data from Jonsson et al) seem quite important to confirm the immature phenotype, but the authors show only a heat map of 10 representative genes in Suppl 8. It would be important to plot all the differentially expressed genes in hPSC-CF vs. haV-CF, especially those highlighted in Fig.7b, 7c and discuss in the text (i.e.TBX20), to determine if the observed differences were also present in the fetal/adult fibroblast comparison. Please re-order the samples uniformly in all the heat-maps.

Treatments to induce differentiation in myofibroblasts with TGF β and Angiotensin II (Fig.8 and Suppl 10): In Fig. 7c hDF panel, the third set of fluorescence and phase images may not come from the same field of view (different orientation of cells), please confirm. In Suppl 10, according to the text, AngII should have no effect on hDF, but the FACS plots show similar values compared to hPSC-CF and hPSC-CF. haV-CF are missing from Suppl 10 and the images used for untreated and OnM hfV-CF and hDF are exactly the same as in Fig. 8. Please substitute the correct images.

REVIEWERS' COMMENTS:

Reviewer #1 (Remarks to the Author):

The authors have replied adequately to the reviewers' comments in their second revision. They have explained more clearly what is the rationale and why it is important to have an “off-the-shelf” source of cardiac fibroblasts (line 46) and they added primary human fetal and adult fibroblasts as control in most of the assays, except for the electrophysiology experiment (Fig.9) were additional tests with fetal fibroblasts couldn't be performed in a “timely fashion”. As for the gene expression, they removed the comparison with Ieda's study (on murine adult and fetal fibroblasts) and referred to a more recent publication from Jonsson et al. on human fibroblasts.

They claim to have found an efficient protocol to obtain CF from hPSCs through SHF progenitors, but all their data show a very immature phenotype and we don't know how these cells compare to the fibroblasts that can be obtained from hPSCs through epicardial progenitors (the experiment was proposed in the first revision and considered beyond the scope of this manuscript). To justify their alternate approach, in the introduction they mention the three main sources of fibroblasts and cite a relatively old paper (13.Cai, C.L. et al. A myocardial lineage derives from *Tbx18* epicardial cells. *Nature* 454,669 104-108 (2008)) to support the idea that just 30% of the fibroblasts are epicardially derived; however the more recent lineage tracing by Moore-Morris T at al, (*JCI* 2014) Fig3 shows that 80% of fibroblasts are epicardial derived and 20% endocardial derived. This should be cited.

Thank you for pointing out this more recent paper by the same authors suggesting a higher percentage of epicardial-derived fibroblasts. We have revised the Introduction to state, “Lineage tracing with the epicardial marker, *Tbx18*, demonstrated *Tbx18*-expressing fibroblasts compromised only one-third of the CFs in embryonic and adult heart¹³; however, a more recent investigation with a *Wtl*-Cre mouse showed up to 80% of CFs in the adult heart were derived from the epicardium¹⁹ .

Specific comments

Regarding the RNAseq comparison in Fig.7: from the dendrograms and the heat maps it looks like the hPSCCF are relatively more similar to haV CF (adult ventricular) than to hPSC, hPSC-derived cardiomyocytes, and dermal fibroblasts. The comparison with fetal fibroblasts (data from Jonsson et al) seem quite important to confirm the immature phenotype, but the authors show only a heat map of 10 representative genes in Suppl 8. It would be important to plot all the differentially expressed genes in hPSC-CF vs. haV-CF, especially those highlighted in Fig.7b, 7c and discuss in the text (i.e.TBX20), to determine if the observed differences were also present in the fetal/adult fibroblast comparison. Please re-order the samples uniformly in all the heat-maps.

Thank you for the comments about the comparison of the gene expression among hPSC-CFs, hfV-CFs and haV-CFs. Yes, the comparison with fetal cardiac fibroblasts (data from Jonsson et al) is important to confirm the embryonic phenotype of the hPSC-CFs. The 10 genes were identified as the representative, most differentially expressed genes in human fetal and adult CFs by the Jonsson's study which is why we chose these 10 genes to compare with our hPSC-CFs gene expression in Suppl Fig. 8. We did analyzed the differentially expressed genes in hPSC-CFs vs. haV-CFs in the revision 2 of the manuscript, and the data were presented in Supplementary

Appendix 1 and 2 (now in our final version of manuscript as Supplementary Data 1 and 2). Among those differentially expressed genes in hPSC-CFs vs. haV-CFs, or vice versa, many are aligned with the differentially expressed genes in human fetal and adult CFs in Jonsson's data such as HEY1, ISL1 and NKX2-5 were significantly expressed in hPSC-CFs or hfV-CF (Jonsson, Appendix 2). On the other hand, GATA6, TBX20 and TCF21 were significantly expressed in haV-CFs in both our data and Jonsson's data.

As per the reviewer's suggestion, we have added the heatmaps comparing gene expression from our hPSC-CFs and haV-CFs with the human fetal and adult CFs of Jonsson et al. for the 57 cardiac factors (Fig. 7b) and the ECM genes (Fig. 7c). These data comparisons are now presented in Supplementary Figure 15.

Regarding the order of the samples in the heatmaps, we did not choose the order of samples, rather it was determined by the statistical analysis (R packages) to generate the dendrograms which are based on the score of unsupervised clustering. We stated this in the Figure legend for clarity such as "...samples of hPSC-CF, haV-CF, hDF, hPSC-CM and hPSC organized by unsupervised cluster analysis."

Treatments to induce differentiation in myofibroblasts with TGFb and Angiotensin II (Fig.8 and Suppl 10): In Fig. 7c hDF panel, the third set of fluorescence and phase images may not come from the same field of view (different orientation of cells), please confirm.

Thank you for your comments regarding Fig. 7c (now Fig. 9 in our final version of manuscript). You are absolutely correct that the phase-contrast images are not the same field of view as the fluorescence images, because the phase images were taken using a different microscope than the fluorescence images to obtain publication quality images from the same biological samples. The phase images aim to show the general characteristics of the cells in culture such as shape, size and confluency and the immunofluorescence focuses on SMA expression. We do not suggest any direct correlation with phase and immunofluorescence. We appreciate the reviewer's observation because it allows us to clarify the images in the Figure legend now stating, "Phase contrast images and SMA immunofluorescence images are from the same biological samples but not the same field of view."

In Suppl 10, according to the text, AngII should have no effect on hDF, but the FACS plots show similar values compared to hPSC-CF and hPSC-CF.

Regarding the AngII experiment results in old Suppl Fig.10 (new Supplementary Figure 18), we suggest that the appropriate comparison is to observe the concentration-response effect for each cell source. So the important comparison is how each cell source responded to AngII. For example, hDF go from 25.2% SMA^{high} to 18.7% and 17.1% in response to 0, 100 and 500 nm AngII (a decrease). In contrast hPSC-CFs go from 15% to 20.4% and 20.3% in response to AngII (an increase). We have clarified our text to note the differential response to AngII as follows, "Ang II treatment induced an increase in the SMA^{high} population as observed with TGFβ1 treatment in both hPSC-CFs and hfV-CFs, but in hDFs Ang II resulted in a decrease in the SMA^{high} population..."

The haV-CF are missing from Suppl 10, and the images used for untreated and 0nM hfV-CF and hDF are exactly the same as in Fig. 8. Please substitute the correct images.

The AngII experiment was to address Reviewer 3's concern to bolster the cardiac specificity of the experiment involving fibroblast activation in hPSC-CFs. We aimed to test if AngII is a more cardiac-specific agonist to stimulate myofibroblast conversion as suggested by the reviewer, so focused more on the hPSC-CFs and hfV-CFs as a control. We were hoping to perform the experiment with the haV-CFs as well, but due to the lack of availability of media to culture the haV-CFs (FGM-3 media from Lonza has been on back-order for a year), we were not able to culture the haV-CFs at that time.

Regarding the images used for untreated and 0 nM hfV-CF and hDF, yes these are the same images in Fig 8 and Suppl Fig. 18 because we performed the TGF β 1 and AngII experiments in parallel using the same starting cells (no agonist).